# Low-frequency ionic-electronic coupling for energy-efficient noise-resilient wireless bioelectronics

Ji Hong Kim[1,8], Haerim Kim[2,5,8], Jaewon Rhee [2,6], Joo Sung Kim[1,7], Hanbin Choi[1], Won Hyuk Choi[1], Yoseph Park[1], Jong Hwi Kim[1], So Young Kim[1], Seungyoung Ahn[2] ✉ & Do Hwan Kim [1,3,4] ✉

Wireless bioelectronics demand transduction strategies that are simultaneously sensitive, noise-resilient, and biologically safe. Conventional wireless sensors typically rely on dielectric capacitors with inherently low capacitance, necessitating operation at MHz frequencies. Such high-frequency coupling often introduces electromagnetic interference, tissue heating, and degraded signal fidelity in biological environments. Here we present a wireless low-frequency electrochemical sensing (WiLECS) platform that couples ionic dynamics with low-frequency LC resonant circuits. The device combines a biocompatible ion gel, composed of a choline-malate ionic liquid embedded in a chitosan matrix with functionalized Au nanoparticles, with a miniaturized LC antenna. Unlike conventional capacitive sensors, WiLECS employs piezo-driven ion redistribution to modulate the dielectric environment of the circuit, enabling sustainable wireless transduction below 1 MHz with high sensitivity and reliability. This approach directly bridges ionic dynamics and electronic resonance, allowing mechanical stimuli to be transduced into biologically safe low-frequency electronic signals. As proof of concept, we demonstrate real-time wireless blood-pressure monitoring in artificial arteries with atherosclerotic plaque, showing resolution of subtle pressure variations under clinically relevant conditions.

Wireless bioelectronics are rapidly transforming the way physiological signals are monitored, enabling untethered, continuous, and real-time data acquisition. At the heart of these systems lies the design of wireless transduction mechanisms, which must balance sensitivity, stability, noise-resilience, and biocompatibility. A variety of approaches, including optical[1–3], acoustic[4–6], and electromagnetic methods[7–9], have been investigated for wireless physiological monitoring, yet each has faced significant challenges in terms of tissue penetration, power efficiency, and signal stability under dynamic biological conditions[10–14]. Among these, inductive coupling has emerged as the most widely adopted strategy, owing to its ability to provide passive and battery-free operation through LC resonators. This architecture has been

[1]Department of Chemical Engineering, Hanyang University, Seoul 04763, Republic of Korea. [2]Cho Chun Shik Graduate School of Mobility, Korea Advanced Institute of Science and Technology, Daejeon 34051, Republic of Korea. [3]Institute of Nano Science and Technology, Hanyang University, Seoul 04763, Republic of Korea. [4]Clean-Energy Research Institute, Hanyang University, Seoul 04763, Republic of Korea. [5]Present address: Samsung Electronics, Hwaseong, Republic of Korea. [6]Present address: Department of Safety Engineering, Chungbuk National University, Cheongju, Republic of Korea. [7]Present address: Thin-Film Device Laboratory, RIKEN, 2-1 Hirosawa, Wako, Saitama 351-0198, Japan. [8]These authors contributed equally: Ji Hong Kim, Haerim Kim. ✉e-mail: sahn@kaist.ac.kr; dhkim76@hanyang.ac.kr

successfully applied in medical implants such as blood pressure sensors[1–9] and cochlear implants[15], highlighting its clinical feasibility and long-term operational reliability.

Nevertheless, conventional inductive systems remain fundamentally constrained by their reliance on dielectric capacitors with inherently limited capacitance. This restriction forces operation in the MHz range, where biological safety concerns, electromagnetic interference, and unstable performance under complex tissue environments emerge as critical challenges[16]. This necessitates MHz-range operation, where electromagnetic exposure has been linked to adverse biological effects, including potential cellular damage, neurochemical imbalances, and oxidative stress[17,18]. Moreover, parasitic interference and unstable capacitance fluctuations under physiological conditions degrade signal fidelity, limiting applicability in bioelectronic platforms. These drawbacks highlight the urgent need to move beyond conventional inductive architectures toward new device-level strategies in which ionic dynamics can directly participate in defining electronic resonance.

Here, we report on a wireless low-frequency electrochemical sensing (WiLECS) platform that establishes such a paradigm shift for energy-efficient, noise-resilient physiological monitoring. Instead of conventional dielectric capacitors, the system integrates a biocompatible ion gel, composed of a choline-malate ionic liquid within a chitosan matrix with functionalized Au nanoparticles, into a flexible LC circuit antenna. This design leverages electric double layer (EDL) formation and piezo-driven ion redistribution to achieve capacitances orders of magnitude higher than dielectric systems, while at the same time enabling ionic transport processes to dynamically tune LC resonance in real time. As a result, mechanical inputs such as pressure are not simply transduced through structural deformation but through ion-electron coupling at the material-device interface, yielding stable and sensitive wireless operation below 1 MHz. To demonstrate the utility of this platform, we implemented WiLECS for continuous, noninvasive blood pressure monitoring in artificial arteries with plaque. By capturing real-time hemodynamic variations under clinically relevant conditions, WiLECS addresses the pressing need for reliable and safe monitoring of cardiovascular diseases, the leading global cause of mortality[19–21]. Beyond blood-pressure sensing, the general strategy of ionic-electronic coupling provides a versatile framework for low-frequency wireless bioelectronics, establishing a new paradigm that bridges ionic transport and electromagnetic resonance for next-generation physiological monitoring.

## Result

### Design of WiLECS for real-time and continuous physiological monitoring

The WiLECS platform (Fig. 1a) is based on an antenna circuit integrated with an ionic capacitive element that directly couples ionic transport with electronic resonance. At the core of this design is a biocompatible ion gel composed of chitosan (CS) polymer blended with choline-malate ([Ch]+[Ma]−) ionic liquids (IL), which establishes electric double layers (EDLs) at the electrode interfaces. To further regulate ion dynamics and enhance interfacial polarization, 2-mercaptoethanol (ME)-functionalized gold nanoparticles (AuNPs) were incorporated into the gel. As illustrated in the side view of the device (Fig. 1b), the ionic capacitive sensor is electrically connected to the antenna via a gold wire, forming a miniaturized LC resonator in which EDL-driven capacitance dynamically tunes resonance frequency. Among the various device components, the ionic capacitive sensor, having the lowest mechanical stiffness, experiences localized pressure changes as the vessel expands and contracts, thereby enhancing sensitivity to blood pressure fluctuations[22]. A distinguishing feature of this architecture is that capacitance modulation does not arise from bulk dielectric deformation but from piezo-driven ionic redistribution at the gel-electrode interface, which couples directly to the electronic resonance

of the antenna. This ion-electron interaction yields exceptionally large capacitance values, enabling stable wireless operation in the sub-MHz (<1 MHz) regime where electromagnetic exposure is biologically safe. While demonstrated here in a cuff-like configuration for monitoring vascular pressure variations, the broader principle is generalizable, as mechanical stimuli that perturb ionic states within the gel can also be transduced into low-frequency electronic signals through the same coupling mechanism.

For wireless signal transmission, the integration of the antenna and sensor forms an inductor-capacitor (LC) circuit, which enables the conversion of ionic signals into resonant frequency shifts in response to pressure variations. The equivalent circuit model of the platform is depicted in Fig. 1c. The resonant frequency, $f_0$, and its shift, $\Delta f_0$, are determined by the antenna's inductance, L, and the sensor's capacitance, C, as described by the following Eq. (1) [7]:

$$f_0 = \frac{1}{2\pi\sqrt{LC}}, \ \Delta f_0 = \frac{\Delta C}{4\pi\sqrt{LC^3}} \tag{1}$$

While conventional capacitive sensors rely on structural deformation of the dielectric under pressure, ion-based capacitive sensors generate a high EDL capacitance as ions migrate to the electrode interface (Fig. 1c, inset). Beyond the absolute capacitance value, the extent of capacitance change in response to pressure is critical, as indicated by the previously introduced equation. To enhance this effect, we functionalized the sensor surface by coordinating AuNPs with the thiol groups of 2-mercaptoethanol (ME) and immobilizing ions at the AuNPs interface through hydrogen bonding between the hydroxyl groups of ME and the [Ch]+[Ma]− ionic liquid (IL) (Fig. 1d and Supplementary Fig. 1). According to the von Mises stress principle, pressure is concentrated at the interface between materials of differing mechanical moduli. While the trapped ions remain immobile under an applied electric field, pressure induces stress at the interface between the ion-gel matrix and the AuNPs, leading to controlled ion release and enhancing pressure-induced capacitance modulation.

Additionally, Supplementary Fig. 2 illustrates the relationship between sensor sensitivity and surface functionalization density. AuNPs with higher surface modification density trap more ions, initially lowering capacitance. However, as pressure is applied, the release of these ions results in a greater capacitance change, thereby improving sensor sensitivity. Overall, the ion gel-based sensor achieves high pressure sensitivity and selectivity while maintaining a high signal-to-noise ratio (SNR), even in the presence of external electrical signals. To establish optimal conditions for maximizing capacitance while maintaining a soft ion gel matrix, we synthesized a series of chitosan (CS)-based ion gels with stepwise increases in IL content. As the ion concentration in the gel increased, a greater number of mobile ions contributed to EDL formation, leading to higher capacitance (Supplementary Fig. 3). Electrochemical impedance spectroscopy (EIS) analysis confirmed that increasing ion concentration enhances ionic conductivity by facilitating greater ion mobility under an applied electric field (Supplementary Fig. 4). Additionally, ions acted as plasticizers within the CS matrix, progressively reducing its modulus. However, when the IL content exceeded 50 wt%, ion leaching occurred, preventing proper gel formation. To overcome this limitation and regulate ion dynamics for mechanotransduction behavior, we introduced functionalized AuNPs as ion trap sites, effectively stabilizing the gel structure while preserving high sensitivity.

As illustrated in Fig. 1e, the WiLECS are constructed through the integration of an ion gel, which imparts high-pressure selectivity, with a double-layer antenna specifically engineered to facilitate low-frequency operation. Most importantly, our WiLECS platform exhibited exceptional sensitivity of 0.0036 mmHg⁻¹ at low resonance frequencies, surpassing the performance of other wireless sensor

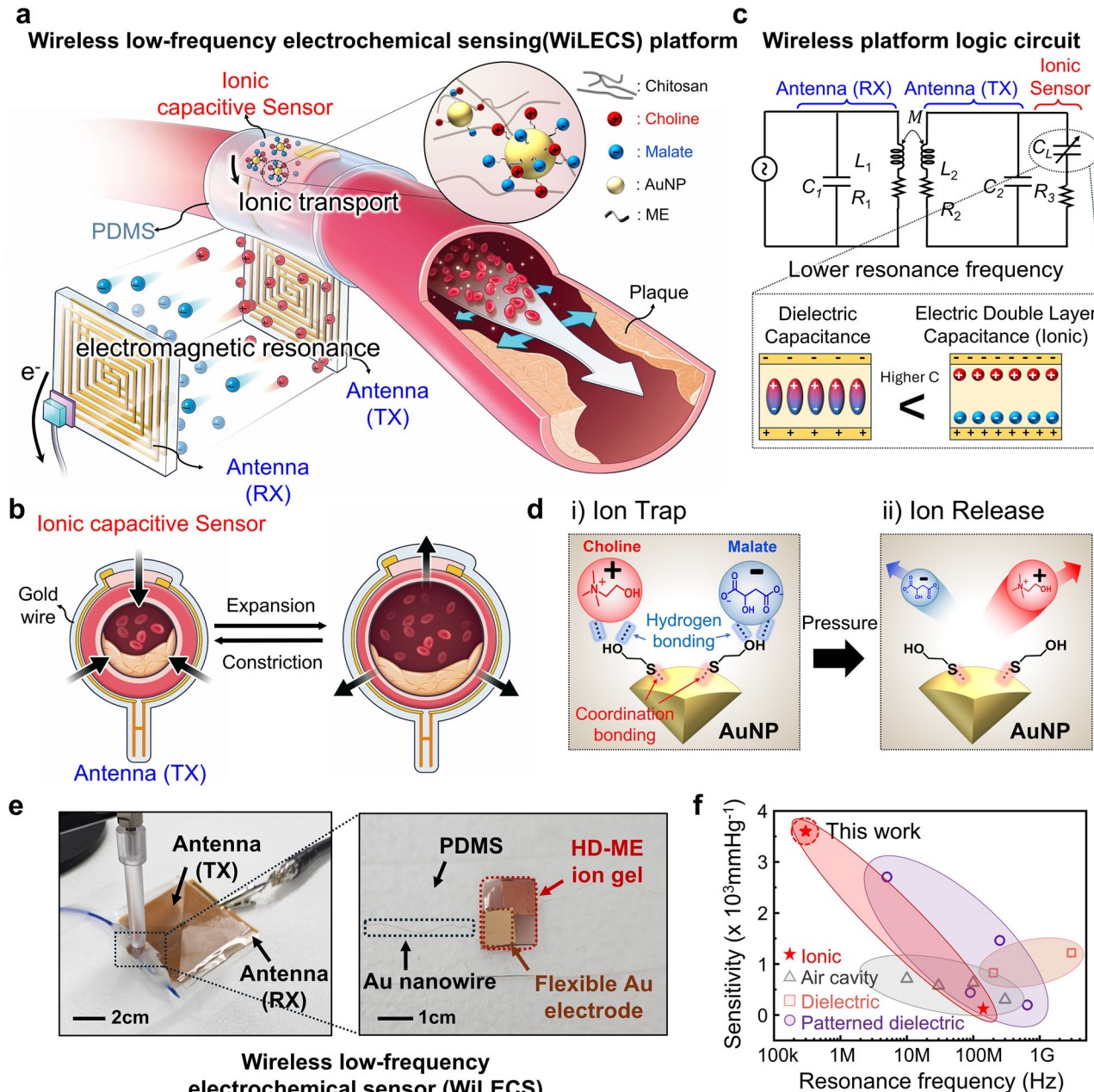

**Fig. 1 | Conceptual design and functional mechanisms of wireless low-frequency electrochemical sensing (WiLECS) platform. a** Illustration of the wireless low-frequency electrochemical sensing (WiLECS) platform. **b** Equivalent circuit design of the LC resonance sensor integration between antennas and ionic sensor (top) and capacitance comparison between dielectric and electric double layer (EDL) capacitance (bottom). **c** Schematic and side-view illustration of arterial wall and device structure change during expansion and constriction. **d** Illustration of a piezo-driven ion confinement mechanism. Before pressure is applied, ions are trapped on the surface of thiol-coordination functionalized gold nanoparticle by hydrogen bonding. After pressure is applied, the trapped ions break the hydrogen bonds and are released, forming an EDL under the influence of the electric field. **e** Photograph of the WiLECS's components. **f** Comparison of the previously reported sensitivity and operating frequency range for wireless pressure sensors.

platforms based on air cavities, dielectric elastomers, and patterned dielectric elastomers (Fig. 1f, Supplementary Table 1)[7,8,23–30].

To confirm the successful surface functionalization of AuNPs, we performed UV-Vis and FT-IR spectroscopy to assess changes in absorbance peaks and bond formation, respectively, resulting from surface modification. As shown in Fig. 2a, untreated AuNPs stabilized in citrate buffer exhibit an absorbance peak at 561 nm. Following surface modification with the ME self-assembled monolayer (SAM) molecules (ME-AuNP), a blue shift in the absorbance peak to 557 nm was observed. This shift originates from electrostatic repulsion between the hydroxyl groups on the functionalized AuNP surface[31]. Further

characterization via FT-IR spectroscopy confirmed successful functionalization. The reaction between the thiol groups of the SAM molecules and AuNPs resulted in the disappearance of the S-H stretching peak at 2556 cm$^{-1}$, with the concomitant formation of an absorption band for the S-Au bond at 2970 cm$^{-1}$ (Fig. 2b). The presence of this S-Au bond provides strong evidence for successful surface modification[32]. Additionally, the CH$_2$ asymmetric peak at 2910 cm$^{-1}$ indicates that the SAM molecules on the AuNP surface are not merely physisorbed but chemically bonded[33]. These results collectively confirm the successful functionalization of AuNPs. The functionalized AuNPs, serving as ion trap sites, were incorporated into the

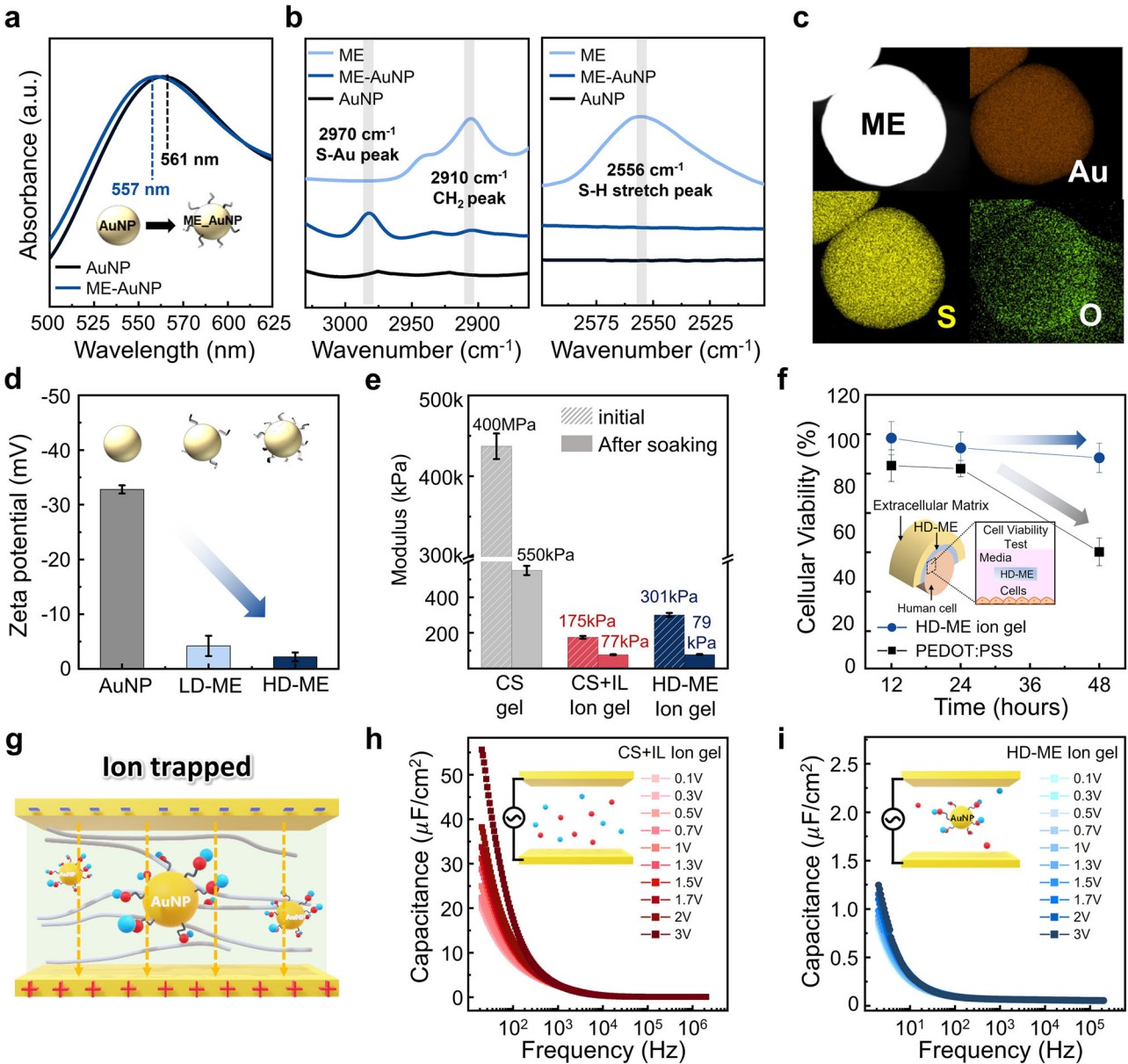

**Fig. 2 | Structural and electrochemical modulation of AuNP-based biocompatible ion gels. a** UV spectra of pristine gold nanoparticles and functionalized gold nanoparticles. **b** FTIR spectra in the spectral regions of 3030-2870 cm⁻¹ (S-Au and CH₂ peak) and 2595-2505 cm⁻¹ (S-H peak). **c** EDS maps for individual elements of Au, S, and O from the functionalized AuNP TEM image. **d** Zeta potential of AuNP, LD-HD, and HD-ME in a citrate buffer. **e** Mechanical property changes after soaking in PBS solution of AuNP, LD-HD, and HD-ME based ion gel. **f** The cellular viability of HD-ME ion gel and the PEDOT: PSS-treated HDFn cells at each time point was normalized to the characterized pure medium-grown cells. **g** Illustration of the role of AuNP. Ions are trapped and limited from migrating without pressure under an electric field. **h, i** Frequency vs. capacitance plots as a function of electric field strength of CS + IL and HD-ME ion gel, respectively.

pristine ion gel (CS + IL, 50 wt% IL), as verified by transmission electron microscopy (TEM) and energy-dispersive X-ray (EDX) analysis (Fig. 2c). EDX analysis revealed a significant concentration of sulfur and oxygen at the AuNP interfaces, confirming that ME molecules were successfully grafted onto the AuNP surface through surface modification.

To optimize ion trapping efficiency, we controlled the surface functionalization density by tuning the AuNP size and adjusting the reaction pH (Supplementary Fig. 5). Smaller AuNPs, even at identical weight fractions, provide a significantly larger surface area than their larger counterparts, facilitating a higher density of surface functional groups and enhancing modification potential. The reaction pH also plays a critical role in AuNP-thiol coordination. The formation of the S-Au bond occurs when the H⁺ ion from the thiol group is displaced, allowing sulfur to bond with gold. However, under low pH

conditions, the high concentration of H⁺ ions in solution impedes thiol deprotonation, thereby hindering S-Au bond formation[34]. On the other side, the surface charge of AuNP formed by H+ induces a repulsive force between AuNPs, and AuNP aggregates in a high pH condition. To balance these effects, ME and AuNPs were reacted at pH 9.1, and the resulting functionalized particles were dispersed at pH 7. Detailed information on the design and synthesis methods can be found in the method section and Supplementary Figs. 6–7. As shown in Fig. 2d, pristine AuNPs in citrate buffer exhibit a high zeta potential due to their negatively charged surface, attributed to the presence of H⁺ ions. Upon surface functionalization, the charged surface area decreases, reducing the zeta potential. Notably, AuNPs functionalized at a higher pH of 9.1 (high-density ME-functionalized AuNPs, HD-ME) exhibit a zeta potential closer to zero, whereas

modification at a lower pH of 3.8 results in low-density surface-functionalized AuNPs (LD-ME).

The synthesized AuNPs were incorporated into the previously developed CS ion gel to fabricate an ionic sensor for wireless pressure monitoring. For effective application in arterial systems, the modulus of the ion gel must be lower than that of the artery, typically around 1 MPa, to minimize mechanical damage. Additionally, as the artery expands during pressure fluctuations, maintaining a lower modulus in the ion gel allows pressure to be more effectively concentrated, ensuring accurate pressure detection. While conventional CS gel exhibits a high modulus of approximately 400 MPa, the incorporation of an ionic liquid (IL) significantly reduces its mechanical stiffness, yielding a modulus of around 175 kPa (Fig. 2e and Supplementary Fig. 8). The HD-ME ion gel, which integrates high-density ME-functionalized AuNPs, demonstrated a slight increase in mechanical properties compared to the original CS + IL ion gel, yet remained well below the modulus of arterial tissue.

Furthermore, when these gels were immersed in phosphate-buffered saline (PBS) to simulate in vivo conditions, their mechanical properties decreased further, enabling enhanced conformal contact with arteries while minimizing potential cellular damage. The HD-ME ion gel exhibited high biocompatibility, maintaining over 80% cell viability for human dermal fibroblasts (HDFn) even after two days of exposure—comparable to the biocompatibility of PEDOT (Fig. 2f). Additionally, the sustained increase in cellular metabolic activity over time suggests that the HD-ME ion gel does not inhibit cell proliferation, further demonstrating its suitability for in vivo applications (Supplementary Fig. 9).

## Multiscale characterization of ion trapping and restricted capacitance dynamics in functionalized ion gels

Raman spectroscopy was performed to confirm whether the ionic liquid remained well-trapped within the ion gel via hydrogen bonding. Choline exists in two conformations, Gauche and Trans, which can be identified by Raman shift peaks at 712 cm$^{-1}$ and ~770 cm$^{-1}$, respectively. The Gauche conformation is favored upon hydrogen bonding[35,36]. As shown in Supplementary Fig. 10, choline interacting via hydrogen bonding with HD-ME exhibits a stronger Gauche conformation peak at 712 cm$^{-1}$ compared to the trans conformation peak at 770 cm$^{-1}$, which is predominant in other materials.

To further investigate the ion-trapping effect of AuNPs under an applied electric field, we fabricated a metal-insulator-metal (MIM) structured device by placing electrodes on both sides of the ion gel and measuring its ionic capacitance (Fig. 2g). As shown in Fig. 2j, the AC voltage bias applied to the sensor was incrementally increased from 0.1 V to 3 V, while the output electric double layer (EDL) capacitance was evaluated across a frequency range of 20 Hz to 2 MHz. In the CS + IL ion gel system, free ions migrate toward the electrodes under an applied electric field, forming an EDL capacitance (Fig. 2h). As the frequency decreases, the extended time available for ion migration enables the formation of a more pronounced EDL at the electrode interface, resulting in higher capacitance values. Additionally, increasing the electric field intensity facilitates greater ion migration, further enhancing capacitance.

In contrast, when HD-ME is incorporated into the ion gel, the AuNPs act as ion trap sites, significantly restricting ion mobility under an electric field. This leads to a substantially lower capacitance compared to conventional ion gels, with minimal variation even as the electric field strength increases (Fig. 2i). To further analyze capacitance behavior across different frequencies, we examined the impact of surface functionalization density by varying the synthetic pH conditions and AuNP sizes. The HD-ME ion gel exhibited greater ion restriction, leading to lower capacitance, whereas the LD-ME ion gel, with reduced surface functionalization, allowed for higher ion mobility and consequently exhibited higher capacitance (Supplementary Figs. 11-12).

## Multimodal characterization of ion trapping in AuNP-integrated ion gels

According to the von Mises stress principle, pressure is concentrated at the interface between materials with differing moduli, particularly between the ion gel matrix and the AuNPs[37]. In this system, ions are confined to the AuNP surface through hydrogen bonding within the HD-ME ion gel, remaining immobile under an electric field; upon pressure application, these hydrogen bonds are disrupted, releasing the ions to migrate under the influence of the electric field (Fig. 3a). To support the hypothesis of piezo-driven ion release, we conducted spectroscopic analysis to examine the hydrogen-bonding strength of the ionic liquid. Before pressure applied, the C-O alcohol peaks of choline and malate appear at 1076 and 1028 cm$^{-1}$, respectively. When pressure is applied, the hydrogen bonds formed with the AuNPs are disrupted, weakening the hydrogen-bonding interactions and resulting in a shift of the C-O alcohol peaks to 1080 and 1033 cm$^{-1}$. This provides evidence that the ionic liquid, which formed hydrogen bonds with the Au nanoparticles, releases ions under applied pressure.

To further support the hypothesis of piezo-driven ion release, we investigated the free ion number density and diffusivity through frequency-dependent dielectric analysis. These parameters were inferred from the frequency dependence of tan δ ($\varepsilon''/\varepsilon'$), specifically by examining the shift in relaxation peaks at the angular frequency ($\omega_{max}$), where complete electrode polarization occurs[38,39]. To experimentally validate this concept, we first fabricated ion gels with varying ionic liquid concentrations and measured their relaxation times. As shown in Supplementary Fig. 13, relaxation time decreases as ionic content increases, likely due to the higher absolute number of free ions present in the system.

Next, to evaluate the effect of ion trapping on free ion concentration, we measured and analyzed the relaxation times of the HD-ME ion gel and the CS + IL ion gel. The HD-ME ion gel exhibited pronounced shifts in relaxation peaks under applied pressure (Fig. 3c). In contrast to the CS + IL ion gel, which showed a 21.1% change in relaxation time, the HD-ME ion gel demonstrated a significantly larger change of 51.8% under applied pressure (Fig. 3d). This behavior suggests that applied pressure increases ion concentration by triggering the dissociation and release of ionic liquids from the AuNP surface, further supporting the role of mechanosensitive ion dynamics in the HD-ME system. Next, we evaluated the sensing performance of various ion-gel-based pressure sensors to investigate the effects of ion trapping and release dynamics across a wide range of pressures. By performing capacitance-frequency sweeps under different pressure conditions (0−45 kPa), we measured the EDL capacitance, which arises from the accumulation of counterions at the ion gel/electrode interface. The sensitivity of the pressure sensor is defined as S = δ(ΔC/C$_0$)/δP, where C$_0$, ΔC, and P denotes the initial capacitance without applied pressure, capacitance change, and applied pressure, respectively. According to this equation, higher sensitivity is achieved with a lower initial capacitance and a greater capacitance change in response to applied pressure. As shown in Fig. 3d and Supplementary Fig. 14, the HD-ME ion gel effectively traps ions, resulting in remarkable sensitivity (S = 17.36 kPa$^{-1}$) under applied pressure. The HD-ME system restricts ion movement under an electric field, leading to a low initial capacitance. When pressure is applied, a large number of trapped ions are released, forming an EDL capacitance and inducing a significant capacitance variation. This substantial capacitance change under pressure enables the device to achieve high sensitivity. As shown in Supplementary Fig. 15, loading-unloading tests were conducted under dynamically increasing applied pressure. At a low pressure of 0.1 kPa, the sensor exhibited a swift response time of 31 ms and a rapid recovery time of 47 ms (Supplementary Fig. 16). Furthermore, continuous loading-unloading cyclic tests confirmed the sensor's reliability and stability, maintaining consistent performance even after 1900 seconds of operation (Supplementary Fig. 17).

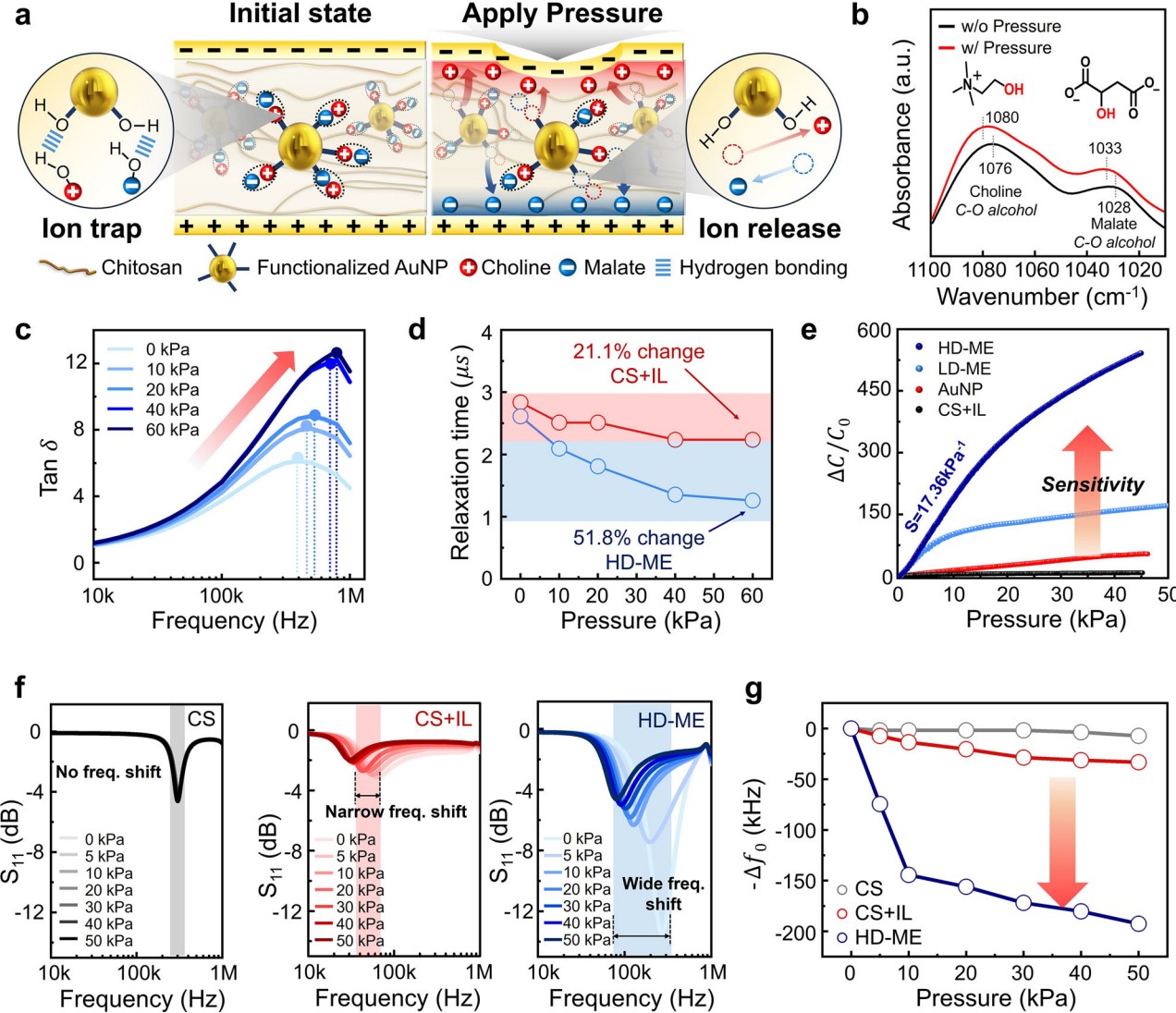

**Fig. 3 | Mechanotransduction mechanism and ion-electron coupled wireless transmission behavior of HD-ME ion gel-based pressure sensor.**
**a** Mechanotransduction of the HD-ME ion gel, where ions are hydrogen-bonded in the pre-stimulus state and released under applied pressure. **b** FTIR spectra of HD-ME ion gel before and after pressure, showing C–O (alcohol) peak shifts of choline and malate due to hydrogen-bond disruption and ion release.
**c** Mechanotransduction ion dynamics of HD-ME ion gel with a stepwise pressure increase. **d** Charge relaxation time decreases with increased pressure input due to

the release of more free ions in the CS + IL and HD-ME ion gel. The HD-ME ion gel exhibits a significant change in ion relaxation time under applied pressure compared to the CS + IL ion gel. **e** Pressure sensitivity comparison among four materials (HD-ME, LD-ME, AuNP, and CS + IL) applied bias of 100 mV at 100 kHz. **f** Wireless resonant frequency sweeps at different pressures of the CS + IL ion gel and HD-ME ion gel, respectively. **g** Resonance frequency shift comparison among three materials (HD-ME, LD-ME, and CS + IL).

## Wireless antenna engineering and resonance analysis for low-frequency, biocompatible pressure sensing

As depicted in Fig. 3e, we integrated pressure sensors based on three different materials (CS gel, CS + IL ion gel, and HD-ME ion gel) with antennas to evaluate the wireless transmission efficiency of the sensors. Considering the potential impact of electromagnetic fields on the human body, we designed the antenna coil to operate within a low-frequency resonance system. To enhance inductance, we increased the number of coil turns within the same spatial area. Additionally, to augment the antenna's capacitance, we engineered a two-layer antenna structure to introduce parasitic capacitance. Furthermore, we employed flexible printed circuit boards (FPCBs), which, due to their thin, flexible nature, offer high capacitance and are easily integrated with the human body. To investigate the sensitivity of the input impedance to applied pressure, we conducted S-parameter simulations. While $Z_{11}$ directly represents the input impedance, $S_{11}$ denotes the reflection coefficient, which quantifies the power reflected due to

impedance mismatches on a logarithmic scale. This logarithmic representation makes $S_{11}$ more suitable for analyzing sensitivity differences, as it effectively highlights small impedance variations on a decibel (dB) scale.

For the conventional CS gel-based dielectric sensor, which lacks an ionic system, the resonance frequency remains unchanged at low frequencies due to its low capacitance and minimal capacitance variation under pressure. The CS + IL ion gel exhibits slight resonance frequency shifts at low frequencies; however, due to its relatively low-pressure sensitivity, the frequency shift remains narrow. While the CS + IL ion gel has a high initial capacitance, resulting in a low initial resonance frequency, its limited capacitance variation under pressure leads to a correspondingly narrow frequency shift. In contrast, the HD-ME ion gel demonstrates a wide resonance frequency shift even at low frequencies, attributed to its mechanotransducing properties and high-pressure sensitivity (Fig. 3f). As a result, increased pressure sensitivity corresponds to a greater shift in resonant frequency, as

illustrated in Fig. 3g. Additionally, analyzing the sensitivity based on the distance between the reader and sensor coils is crucial for practical implementation. To assess measurement sensitivity at varying coil distances, we performed S-parameter simulations using the initial capacitance values of the two pressure-sensitive materials and the coil's electrical parameters. The gap distance was varied from 1 mm to 25 mm, accounting for the maximum implantation depth under consideration.

To accurately capture changes in magnetic coupling due to coil separation, we incorporated experimentally measured coupling coefficients into the simulations. The coupling coefficient (k), which quantifies the degree of magnetic coupling between the two coils, is defined using the mutual inductance (M) and the self-inductances of the two coils ($L_1$, $L_2$) according to the following Eq. (2):

$$k = \frac{M}{\sqrt{L_1 L_2}} \qquad (2)$$

As shown in Supplementary Fig. 18 and Table 2, the measurement sensitivity as a function of the gap between the sensor coil and the reader coil was evaluated using S-parameter analysis. The results indicate that the HD-ME ion gel exhibits superior sensitivity across varying gap distances compared to the CS + IL ion gel. Consequently, the HD-ME-based wireless sensor demonstrated high-sensitivity wireless transmission even at a distance of 2.5 cm, which corresponds to the typical depth of arteries. Furthermore, as depicted in Supplementary Fig. 19 and Table 3, the sensor maintained reliable operation even when utilizing a significantly smaller antenna size (1.25 cm × 1.25 cm), confirming its feasibility for miniaturized, implantable applications.

### Demonstration of WiLECS in artificial artery models for wireless blood pressure monitoring

We further demonstrated a WiLECS platform for detecting blood pressure variations in an artificial artery system (Fig. 4a). The HD-ME ion gel was placed into the grooves of a cuff-shaped PDMS mold, with flexible electrodes (Cr/Au deposited on PET) positioned above and below the ion gel. This ionic pressure sensor, connected to an antenna via Au nanowires, was integrated within a 6.0 mm-diameter balloon catheter and inflator, simulating arterial expansion and constriction. To model plaque-induced blood flow restriction, fat was injected into the balloon catheter. The balloon expanded and contracted due to air injection from the inflator, cyclically applying and releasing pressure on the sensor. The resonant frequency of the device, which links the ionic sensor to the TX antenna, enables WiLECS with the external RX antenna. The resonant frequency shifts in response to applied pressure, allowing real-time measurement of blood pressure variations. As illustrated in Fig. 4b, plaque buildup inside the artery leads to an increase in vessel diameter, ultimately causing elevated blood pressure.

To validate the feasibility of blood pressure monitoring, we integrated the capacitive sensor with the artificial artery system. The expansion of the artificial artery exerted mechanical pressure on the gel, leading to capacitance variations. Figure 4c presents the capacitance variations of sensors based on three different materials (CS gel, CS + IL ion gel, and HD-ME ion gel) in both normal and oily artery models. These measurements enabled a comparative assessment of sensor performance under varying arterial conditions. The HD-ME ion gel-based sensor exhibited significantly higher sensitivity than both the CS gel and CS + IL ion gel, demonstrating its superior capability in detecting elevated arterial pressure caused by plaque formation.

Next, as shown in Fig. 4d, the wireless ionic sensor was integrated with the artificial artery system, and frequency shifts corresponding to blood pressure variations were successfully measured (Supplementary Movie 1). Similar to the capacitive sensor results, the HD-ME ion gel-

based device achieved much higher sensitivity than the other materials, confirming its effectiveness for accurate blood pressure monitoring and highly sensitive wireless transmission, enabled by its ion trap-and-release mechanism. Based on the previously conducted artificial blood pressure measurements, we evaluated the signal-to-noise ratio (SNR) for both the capacitive ionic sensor and the wireless ionic sensor across different materials. As shown in Supplementary Fig. 20, the HD-ME-based sensor platform precisely measured signals only when pressure was applied, leveraging the ion trap-and-release effect. Furthermore, it demonstrated an SNR of 78 dB, which is over four times higher than that of the conventional CS polymer gel-based dielectric sensor (15 dB).

Additionally, to verify the feasibility of wireless transmission, we evaluated transmission efficiency with a spacer (Fig. 4e, Supplementary Movie 2) and porcine skin (Fig. 4g, Supplementary Movie 3) inserted between the transmitting and receiving antennas. As a result, when comparing performance with a 5 mm spacer (Fig. 4f) and 7 mm porcine skin (Fig. 4h), the wireless signal decreased from −13.4 dB to −10.26 dB and −4.53 dB, respectively, yet reliable pressure readout was maintained. Notably, by operating in the bio-safe low-frequency regime, the WiLECS platform highlights the power of ion-electron coupling to enable sensitive, stable, and sustainable wireless physiological monitoring.

## Discussion

In conclusion, we establish a wireless bioelectronic platform that directly couples ionic dynamics with electronic resonance, offering a safe and sustainable alternative to conventional inductive coupling. By integrating an ionic mechanosensitive HD-ME ion gel into custom-designed LC circuits, giant EDL capacitances dynamically modulate resonance, enabling stable operation below 1 MHz. The HD-ME ion gel itself provides exceptional mechanical compliance and pressure-selective responsiveness, ensuring intimate integration with soft tissues and preserving a high signal-to-noise ratio (SNR). Through this synergistic material–device interaction, mechanical inputs are transduced not merely through structural deformation but via ion–electron coupling at the gel-electrode interface, enhancing sensitivity and reliability.

While demonstrated here in vascular sensing, this approach represents a generalizable framework for bio-integrated wireless systems. The combination of ion-electron coupling and mechanosensitive ion gels provides enhanced signal fidelity, improved energy efficiency, and intrinsic compatibility with dynamic biological environments. Beyond continuous blood pressure monitoring, this paradigm lays the foundation for autonomous, real-time, and closed-loop bioelectronic technologies, redefining the architecture of wireless communication in implantable devices and advancing toward long-lived, bio-safe, and sustainable healthcare systems.

## Methods

### Synthesis of choline-malate [Ch]⁺[MA]⁻ ionic liquid

Choline-Malate ([Ch]⁺[MA]⁻) ionic liquid was synthesized via salt metathesis. A 2 M solution of malic acid in methanol was gradually introduced to a 0.8 M solution of choline bicarbonate (Sigma-Aldrich; St. Louis, MO, USA) under gentle stirring at 25 °C for 4 h to allow the release of carbon dioxide. Methanol and water were removed through rotary evaporation at 80 °C for 12 h, followed by drying in a vacuum oven at 80 °C for 48 h.

### Biocompatible ion gel fabrication

Biocompatible ion gel was prepared in three main steps: (i) preparation of 2-mercaptoethanol (ME) functionalized Au nanoparticle; (ii) preparation of chitosan-based ion gel; (iii) Integration of functionalized Au nanoparticles and ion gel-component solution, followed by an optimized heat-treatment process to develop ME ion gel films. In the first step, 556 μL of the 2-mercaptoethanol (≥ 99 %, Sigma-Aldrich, Saint

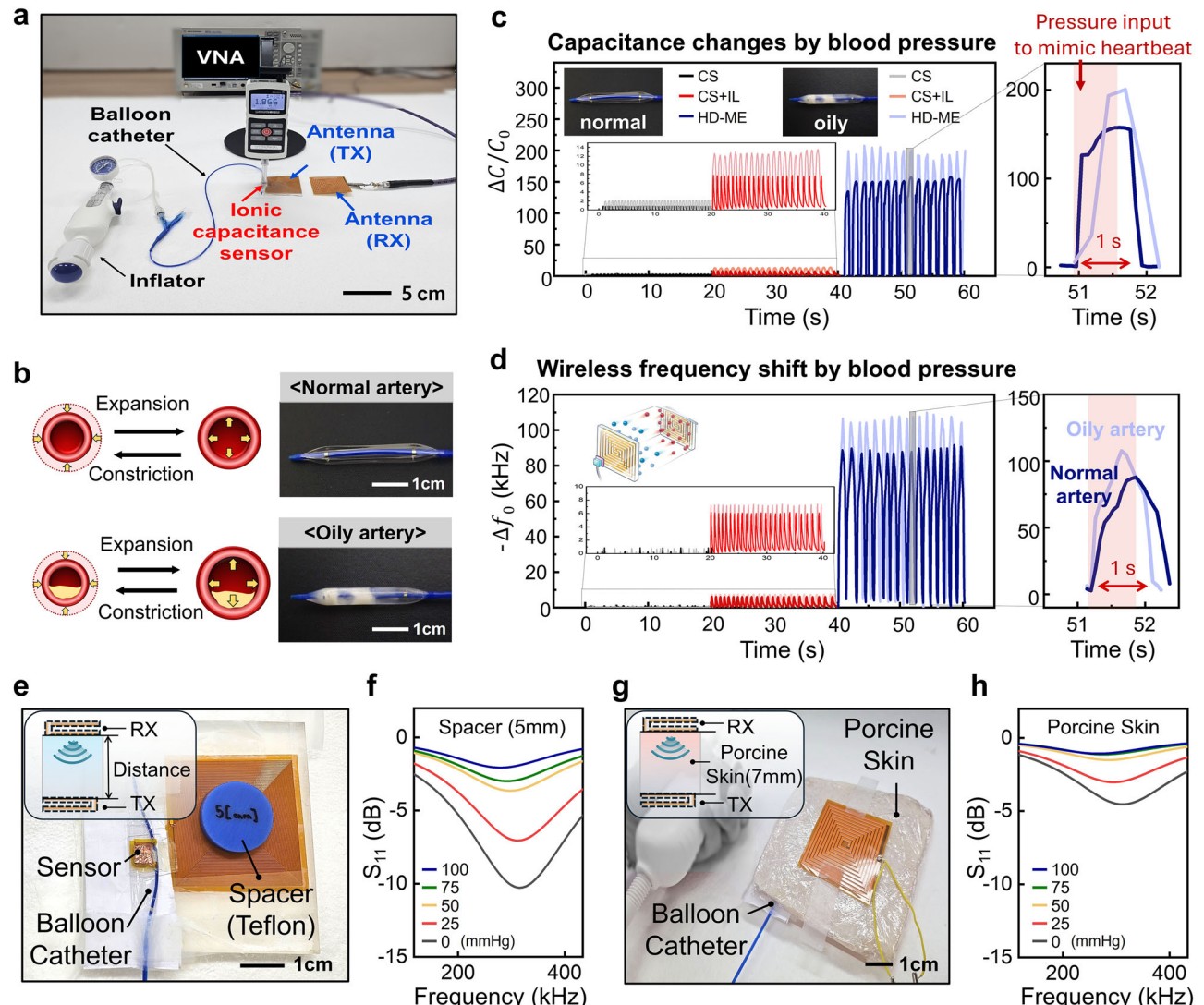

**Fig. 4 | Demonstration of wireless sensing between normal and oily artery conditions. a** Photographs of the wireless pressure sensor platform integrated with balloon catheter and inflator. **b** Schematic illustration and photograph of a normal and oily artery model's blood pressure dynamics. **c, d** Relative capacitance change plots and wireless pressure sensing comparison between normal and oily artery as a types of ion gel. Inset shows the capacitance changes and resonance frequency changes between artery expansion and constriction. **e** Photograph of the balloon catheter integrated with WiLECS, incorporating a spacer between the RX and TX units. **f** Resonance frequency shifts as a function of catheter pressure, measured in a configuration with a spacer inserted between the TX and RX antennas. **g** Photograph of the balloon catheter integrated with WiLECS, incorporating a porcine skin between the RX and TX units. **h** Resonance frequency shifts as a function of catheter pressure, measured in a configuration with a porcine skin inserted between the TX and RX antennas.

Louis, MO, USA) was added to 5 ml of the AuNPs solution with pH buffer (50 nm, stabilized suspension in various buffer, Sigma-Aldrich, Saint Louis, MO, USA) with pH buffer (pH 3.8, 4.5, and 9.1). The mixed solution was subsequently sonicated in an ultrasonic bath using a glass vial for 30 mins and then incubated for 12 h at room temperature under a slow vortex. Excess thiols in the solution were removed by twice centrifugation for 12 mins (at 12,500 g, 4 °C). After purification, the thiol-functionalized AuNP were redissolved in 1 ml of Millipore-Q water. In the second step, ion gel solution was prepared by mixing chitosan solution, which includes 2 wt% of chitosan (medium molecular weight, Sigma-Aldrich, Saint Louision, MO, USA) in 1 v/v% acetic acid (99.7%, DAEJUNG, Korea) and synthesized [Ch]⁺[MA]⁻. Depending on the concentration (10, 30, 50 wt%) of ionic liquid, [Ch]⁺[MA]⁻ was mixed with chitosan solution for 24 h on room temperature, followed by sonicating it for 30 min for homogeneous suspension of ionic liquid in the solution. In the last step, 1 ml of synthesized AuNPs solution was mixed with the ion gel solution and blended with mild stirring overnight. 10 g of the ion gel solution was poured into a glass petri dish and

cured for 48 h at room temperature. The fabricated film was peeled off from the substrate to obtain a biocompatible ion gel film

### Wireless ionic sensor fabrication

The PDMS solution, which has a spacer layer (1 cm by 1 cm, Thickness: 500 $\mu m$), was prepared with 1:10 weight ratio of PDMS and curing agent. The solution was annealed at 80 °C for 4 h in the convention oven. The spacer incorporates a capacitive ionic sensor with an MIM structure. Top and bottom gold electrodes (3 nm Cr/ 30 nm Au) for capacitive ionic sensors have been deposited on PET substrate by thermal evaporation and patterned via metal mask. Gold wires were attached between the electrodes and antennas to connect the circuit. The completed wireless ionic sensor wraps around the balloon catheter in a cuff configuration, measuring pressure through balloon expansion. An artery model was formed with PTA balloon catheter (GENOSS, GPNC-06-040-050) connected to inflator (GENOSS, GBI-B20V). To create the oily catheter, liquid lard was injected into the catheter and solidified at room temperature.

## Material characterization

The optical absorbance of the Au nanoparticles was analyzed using UV-vis-NIR spectroscopy (Jasco, V-770). FTIR spectra were collected with a Bruker Optics GmbH (Germany) spectrometer operating in attenuated total reflection (ATR) mode using a ZnSe crystal. Each spectrum was averaged over 256 scans with a resolution of $1\,cm^{-1}$, covering the range of 4000 to $450\,cm^{-1}$. FTIR peak deconvolution was carried out using Gaussian peak fitting through iterative optimization for best-fit results. The surface morphology of the Au nanoparticle films was characterized using field-emission scanning electron microscopy (FE-SEM, JSM-6700F, JEOL) equipped with EDS system for elemental analysis. Raman spectra were obtained with a Thermo Scientific™ DXR™2 Raman microscope and the DXR™2xi Raman imaging system. During Raman imaging, a 532 nm laser with a power of 2.0 mW was used, providing a spot size of approximately $10\,\mu m^2$ and a spatial resolution of 3 μm. Zeta potential analysis was performed using a Malvern Zetasizer NanoZS in backscatter mode, with a working voltage of 40 V applied during measurements.

## Mechanical property

A universal testing machine (UTM QRUTS-S105, QURO) equipped with a 1-kN load cell was used to evaluate the mechanical properties of all polymer films at 25 °C, applying a stretching speed of 10 mm/min. The samples were prepared and tested following ASTM standards (Test Method D 638-02a, specimen type V). Young's modulus (Y) was determined from the slope of the stress-strain curves within the 0–5% strain range. The mechanical properties reported in this study are expressed as the mean ± standard deviation, based on five samples tested under identical experimental conditions.

## Cell viability test

Cells were cultured in growth medium and treated with the PEDOT:PSS and HD-ME ion gel (1.5 mg) for 12, 24, and 48 h. After exposure, 10% MTT solution (Sigma-Aldrich) in growth medium was added, and the cells were incubated at 37 °C with 5% $CO_2$ for 4 h. To terminate the reaction, a lysis buffer consisting of 50% (w/v) N,N-dimethylformamide (DMF, Sigma-Aldrich) and 10% (w/v) sodium dodecyl sulfate (SDS, Sigma-Aldrich) in distilled water was introduced into each well. The 96-well plates were kept at room temperature in the dark for 2 h to allow formazan crystals to diffuse into the medium. Optical density (OD) was measured at 570 nm using a microplate reader (Thermo Fisher Scientific).

## Electrical characterization

Electrochemical impedance spectroscopy (EIS) is an essential method for investigating ion transport phenomena within polymer electrolytes and their interfaces, such as electrode–electrolyte interactions. EIS measurements were carried out at room temperature using a PGSTAT302N electrochemical analyzer (Metrohm Autolab) over a frequency range of 0.1 Hz to 100 kHz with a 10 mV AC signal. A coin cell assembly from Hohsen Corp. (Japan) was utilized to conduct EIS tests on various polymer films under different conditions, including with and without applied pressure. In these measurements, polymer films with a thickness of approximately 500 μm were positioned between two stainless steel discs (10 mm in diameter, serving as electrodes) to create a piezocapacitive device configuration. The impedance spectra were analyzed and fitted with appropriate equivalent circuit models using NOVA software (Metrohm Autolab) to determine the bulk resistance (Rb) of the devices. Ionic conductivity (σ) was calculated from the bulk resistance according to the equation:

$$\sigma = \left( \frac{l}{R_b \times A} \right)$$

where σ is the ionic conductivity, $l$ is the thickness of the polymer film between the electrodes, $A$ is the electrode area, and $R_b$ is the bulk resistance derived from the Nyquist plots. Capacitance measurements were performed at room temperature using an Agilent E4980A Precision LCR Meter. Piezocapacitive devices were fabricated by sandwiching the ion gel films (film thickness ~ 500 μm, film area $0.75\,cm^2$) between two Au electrodes (surface resistance ~ $10\,\Omega\,sq^{-1}$). The gold wires (Nilaco Corp., diameter: 50 μm) were attached to the electrodes for connections with the measuring instrument.

## Analysis of pressure response and sensitivity

A custom-designed sensor-probe station featuring a programmable stage with xy- and z-axis control (0.1-μm resolution) and a force gauge (Mark-10, 0.005-N resolution) was utilized to evaluate the pressure response of the assembled sensors. The applied pressure was determined by dividing the applied load by the sensing area of the unit device or pixel. The measurement setup was integrated with a LabVIEW-based program, enabling real-time recording of both the capacitance change and applied load. Pressure sensitivity across different pressure ranges was calculated from the slope of the plot showing the relative change in capacitance versus pressure.

## Wireless platform characterization

The final proposed system consists of an antenna with two spiral coil structures and a material whose capacitance varies depending on pressure. In order to identify the antenna design and the inductance and coupling coefficient of the proposed system, ANSYS HFSS, a finite element method-based simulation, was used to model the antenna. Using the electrical parameters of the antenna extracted through EM simulation, a network was configured with Keysight Advanced Design System (ADS) to verify the reflection characteristics ($S_{11}$). The selected dielectric material of the antenna was polyamide, turns were made of copper, and the leader coil was manufactured with 20 turns and the sensor coil with 70 turns. In order to verify the measurement of the proposed system, the reflection coefficient according to pressure was measured using a vector network analyzer (VNA), which is a test equipment that measures the parameters of a network by sweeping the frequency. The VNA used is Keysight E5071C (9 kHz to 8.5 GHz), and the measurement frequency range is 10 kHz to 1 MHz.

## Data availability

All relevant data supporting the results of this study are available within the article and its supplementary information files. Further data is available from the corresponding authors upon request. Source data are provided with this paper.

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

## Acknowledgements

This work was supported by the National Research Foundation of Korea(NRF) grant funded by the Korea government(MSIT) (RS-2025-00515479, No. RS-2024-00405818), the Pioneer Research Center Program through the National Research Foundation of Korea funded by the Ministry of Science, ICT & Future Planning (RS-2022-NR067540). This research was supported by Korea Basic Science Institute (National research Facilities and Equipment Center) grant funded by the Ministry of Education (RS-2024-00436346). This work was partly supported by Institute of Information & communications Technology Planning & Evaluation (IITP) grant funded by the Korea government(MSIT) (No.RS-2020-II200839, Development of Advanced Power and Signal EMC Technologies for Hyper-connected E-Vehicle).

## Author contributions

S.A. and D.H.K. supervised the project. D.H.K., S.A., Ji Hong Kim, and H.K. developed the theoretical concepts and designed the experiments. Ji Hong Kim and H.K. carried out all the experiments. J.R., J.S.K., H.C., W.H.C., Y.P., Jong Hwi Kim, and S.Y.K. helped with experimental procedures. D.H.K. and S.A. reviewed and provided feedback on the manuscript. D.H.K., S.A., Ji Hong Kim, and H.K. wrote the manuscript, with D.H.K. responsible for revisions. All authors discussed the results and contributed to the final manuscript.

## Competing interests

The authors declare no competing interests.
