## [Transparent Peer Review file · Nature Communications]

Low-frequency ionic-electronic coupling for energy-efficient noise-resilient wireless bioelectronics

Corresponding Author: Professor Do Hwan Kim

Version 0:

Reviewer comments:

Reviewer #1

(Remarks to the Author)

The work by Kim et al. introduces a wireless low-frequency electrochemical sensing (WiLECS) platform. Notably, these biocompatible ion gel-based sensors surpass traditional dielectric counterparts, offering distinct advantages in low-frequency (< 1 MHz) sensing, high sensitivity, and reliability. The practical potential of the HD-ME-based platform was illustrated by monitoring blood pressure in artificial arteries. Although the study demonstrates significant novelty in material design, comprehensive experimental data is currently lacking. To strengthen the validity of the conclusions, it is essential for the authors to provide convincing experimental data. Please be advised that formal acceptance depends on the rigorous completion of these additional experiments.

1. In Figure 1e, does the small diameter of the Au nanowires affect resistance? Additionally, is the smaller Au electrode designed to mitigate edge effects or enhance mechanical stability?
2. Ion gels with high ionic liquid content are often prone to liquid exudation (syneresis) under mechanical compression. Specifically, regarding the functionalized Au nanoparticles (AuNPs) embedded in the matrix, is there any risk of such particles migrating towards the electrodes or device edges along with the ionic liquid under high pressure? Could the authors provide evidence (e.g., optical or SEM images after compression) to prove that the AuNPs are firmly anchored within the chitosan matrix and do not cause interfacial contamination or short circuits?
3. The specific caption 'Mechanotransduction ion dynamics' is somewhat vague for Figure 3c. Since the plot directly displays the loss tangent $\tan \delta$ spectra, it would be more accurate to label it as 'Frequency dependence of dielectric loss $\tan \delta$ under different pressures'. This clearly describes the observable data, while the text can discuss how this reflects the underlying ion dynamics.
4. The decrease in relaxation time with increasing pressure in Figure 3d appears to contradict the expectation that compression usually hinders ion transport due to reduced free volume. Please provide a more fundamental explanation: is this result an artifact of reduced contact resistance, or does it reflect a genuine change in the ionic hopping mechanism?
5. Regarding Figure 3f, I noticed that the 3 dB bandwidth for the CS+IL and HD-ME samples increases significantly with rising pressure, suggesting a degradation in the Quality Factor (Q). I am interested to know how the Q factor evolves under pressure for these groups. Furthermore, considering this trade-off between spectral sharpness and sensitivity, what is the primary advantage of the HD-ME configuration that justifies its selection as the optimal material?
6. The schematic illustration of the hydrogen bonding mechanism in Figure 3a is chemically inaccurate. Please correct this to show the appropriate donor-acceptor interaction (H...O).
7. In Figure 4, there are several figures with no scale bar, such as panel 4a, the upper right picture of 4b, 4e and 4g. The figures contain non-standard notations, such as 'S11', 'cm-1' (fig. 2b), which lack the necessary subscripts and superscripts.
8. While the manuscript notes low-frequency stability, Figure S15 shows capacitance drift at 30 kPa. Please define the specific stable pressure range and clarify if the wireless frequency readout exhibits similar drift under this pressure.
9. Real blood vessels are composed of multilayered biological tissues within an ion-rich physiological environment, which differs fundamentally from the synthetic model. This discrepancy may introduce background noise or signal attenuation, thereby affecting the SNR. To ensure clinical relevance, the authors are advised to validate the device performance using real animal blood vessels.
10. Please specify the exact pressure applied to achieve the 78 dB SNR in Figure S20. Providing the corresponding frequency signal and noise floor data would greatly enhance the completeness and transparency of this characterization.

Reviewer #2

(Remarks to the Author)

This manuscript presents a soft, wireless LC pressure sensor (WiLECS) in which a mechanosensitive ion gel converts arterial pressure into large, low-frequency capacitance modulation through a hydrogen-bond-mediated trap-and-release mechanism. The mechanisms of mechanically gated ionic trapping/release and resonance-frequency tuning are elucidated through systematic analyses, along with demonstrations of wireless LC behavior in arterial phantom models. This is a highly relevant and compelling topic in the fields of iontronics and biointegrated cardiovascular monitoring, and the work could provide an even deeper mechanistic understanding for researchers and readers. I recommend that the manuscript be considered for publication in Nature Communications after minor revisions addressing the detailed comments below.

1) The main concept appears to be a wireless LC pressure sensor with an ionic gel capacitor, Science Advances 11(11) (2025), eadu6086, has demonstrated polyelectrolyte-based wireless iontronic sensors using an ionic medium. Could the authors more clearly clarify the conceptual and practical novelty of WiLECS relative to both conventional elastomeric LC sensors and this recent ionic/iontronic work, and also include the device from Sci. Adv. 11(11), eadu6086 in the Figure. 1f figure-of-merit comparison for a quantitative benchmark?

2) For practical use in wearable or implantable settings, environmental stability (thermal and electrochemical) is critically important for reliable device operation. I therefore encourage the authors to more explicitly address the stability of the device/gel under relevant conditions (e.g., body temperature, and long-term bias etc), or at least to discuss these aspects and the expected failure modes and limitations.

3) The authors attribute the enhanced sensitivity to a “trap-and-release” mechanism at ME-functionalized AuNPs. However, the observed FTIR shifts and relaxation-time changes could also be interpreted as generic pressure-induced changes in ion distribution. Can the authors more convincingly argue that ion trapping at AuNPs is essential, rather than simple EDL compression?

4) In the current manuscript, the main stated advantage of using Au nanoparticles is their biocompatibility, but the specific benefits of Au compared with other possible fillers (e.g., in terms of interfacial chemistry, ionic trapping, mechanical properties, or electrical performance) are not clearly articulated. I encourage the authors to clarify why Au was chosen and what unique role it plays in the gel design beyond biocompatibility.

Reviewer #3

(Remarks to the Author)

The article entitled “Low-frequency ionic-electronic coupling for energy-efficient noise-resilient wireless bioelectronics” by Kim et al. introduces an approach for wireless electro-chemical sensing that combines ionic dynamics with low-frequency LC circuits. The proposed material and concept would be useful for the development of associated technologies, but some of the below comments should be addressed.

1. Explain the selection of 50 wt.% IL as the primary composition (Authors claim that >50% causes leaching). To support the choice of 50 weight percent, provide a brief analysis of the trade-off IL% vs. modulus vs. capacitance vs. leaching.
2. The sample names (HD-ME, LD-ME, CS+IL, pristine CS) aren't always used consistently throughout the paper. Please keep the name uniform throughout to avoid confusion.
3. Fabricate identical gels with AuNPs that are not functionalized (pristine AuNP) and with ME alone (no AuNP) to show the necessity of both ME-AuNP coordination and AuNP mechanical contrast for the trap-release behavior. Some comparisons exist (they show AuNP and LD-ME), but a complete control set (no AuNP, pristine AuNP, ME-only) would be helpful.
4. Given that the sensing mechanism depends on the trapping and release of ions from the AuNP surfaces, it would be crucial to determine the stability of this behavior over time. The SI shows short-term durability and 48-hour cell viability, but there's no information on ion or IL leaching, or how the device performs over several days or weeks in physiological conditions (37 °C saline). Since these are crucial for evaluating implantable use, I suggest including accelerated leaching tests and multi-day wireless stability measurements (baseline drift and sensitivity after few days gap).
5. The main idea of the paper is that higher sensitivity and SNR result from starting with a low initial capacitance from trapped ions and then obtaining a large ΔC under pressure. To help readers fully see this relationship, it would be useful to include a simple quantitative comparison, maybe a table showing the initial C, ΔC , Δf , and SNR for each material measured under the same conditions.
6. The manuscript highlights the advantage of using sub-MHz frequencies for bio-safe operation, but actual safety depends on factors like specific absorption rate, induced electric fields, and any tissue heating. Including some measurements or simulations that illustrate the field strength or temperature change for your operating conditions and distances would be very beneficial. If the SNR is actually very low, a brief explanation of safe operating limits would be helpful; if not, data would support the claim.

Version 1:

Reviewer comments:

Reviewer #1

(Remarks to the Author)

I think the revisions have substantially improve the quality of the manuscript. I am supportive to the publication at its current

form.

Reviewer #2

(Remarks to the Author)

The authors addressed the questions, the manuscript is ready for publication.

Reviewer #3

(Remarks to the Author)

I believe that the revisions address the points that were raised in the original review. The paper is now suitable to be accepted in nature communications.

Reply to Reviewer #1

Thank you for your invaluable comments. We fully revised the manuscript according to your comments.

Overall Comment: *The work by Kim et al. introduces a wireless low-frequency electrochemical sensing (WiLECS) platform. Notably, these biocompatible ion gel-based sensors surpass traditional dielectric counterparts, offering distinct advantages in low-frequency (< 1 MHz) sensing, high sensitivity, and reliability. The practical potential of the HD-ME-based platform was illustrated by monitoring blood pressure in artificial arteries. Although the study demonstrates significant novelty in material design, comprehensive experimental data is currently lacking. To strengthen the validity of the conclusions, it is essential for the authors to provide convincing experimental data. Please be advised that formal acceptance depends on the rigorous completion of these additional experiments.*

Response: We sincerely thank Reviewer 1 for the positive assessment and for recognizing the novelty of our work. We have carefully addressed all comments, and these revisions have significantly improved the quality of the original manuscript. We hope that Reviewer 1 finds the revised version satisfactory.

Q1) *In Figure 1e, does the small diameter of the Au nanowires affect resistance? Additionally, is the smaller Au electrode designed to mitigate edge effects or enhance mechanical stability?*

Response: We appreciate the reviewer for this insightful question. Although each Au nanowire has a small diameter, the electrode forms a dense percolating network, and the overall series resistance of the Au nanowire layer is much smaller than the ionic impedance of the device in the measured frequency range, so it does not limit the device performance. The smaller flexible Au electrode is intentionally designed to be slightly smaller than the active gel region. This geometry reduces stress concentration at the electrode edge, which improves mechanical robustness under repeated loading, and helps confine the effective electric field to a mechanically uniform central region, thereby minimizing edge-related artifacts in the electrical response. Additionally, in implantable applications, where the electrode is encapsulated in PDMS to prevent interaction with the surrounding ECM ions, the smaller diameter of the Au nanowires allows for easier and more efficient encapsulation, contributing to the long-term stability of the device. To highlight this important progress, we have added the following sentence **on page 6, line 4-10 in the revised manuscript**. Additionally, to address the reviewer's concerns and enhance the readers' understanding, we have included further explanations not only about the gold nanowires but also about the overall content of **Figure 1e**.

"The smaller flexible Au electrode and Au nanowire, with their reduced diameter, form a dense percolating network that ensures low series resistance, without limiting device performance. This design reduces edge stress and improves mechanical stability under repeated loading,

while maintaining a uniform electric field. In addition, in implantable applications, the smaller Au nanowires facilitate efficient PDMS encapsulation, enhancing long-term stability by preventing interactions with surrounding ECM ions.”

Q2) *Ion gels with high ionic liquid content are often prone to liquid exudation (syneresis) under mechanical compression. Specifically, regarding the functionalized Au nanoparticles (AuNPs) embedded in the matrix, is there any risk of such particles migrating towards the electrodes or device edges along with the ionic liquid under high pressure? Could the authors provide evidence (e.g., optical or SEM images after compression) to prove that the AuNPs are firmly anchored within the chitosan matrix and do not cause interfacial contamination or short circuits?*

Response: We sincerely appreciate the reviewer’s thorough and insightful comments. As the reviewer rightly pointed out, ion leakage can indeed occur depending on the ion content and the external stimuli applied to the system. This phenomenon originates when the ions within the ion gel do not sufficiently interact or bond with the polymer matrix or the functionalized Au nanoparticles, resulting in poor compatibility and potential ion migration out of the matrix. Such ion leakage is a known and intrinsic challenge in ion-gel-based systems. To effectively address this issue, we incorporated functionalized gold nanoparticles into the chitosan matrix and systematically optimized the ionic liquid (IL) content to achieve a stable balance between ionic mobility and matrix confinement. Because our material is specifically designed for pressure-sensing applications, it is essential that ion leakage is completely suppressed both before and after the application of mechanical pressure. Through careful optimization, we determined that an IL content of 50 wt% relative to chitosan provides the optimal composition. At this ratio, the ions are sufficiently immobilized through strong interactions with the polymer matrix and the functionalized Au nanoparticles, ensuring that no ion leakage is observed even under applied pressure.

As the reviewer suggested, we used optical microscopy to observe ion leakage both before and after pressure application, and these findings are now provided in **Supplementary Figure 8** for readers to review. To highlight this important progress, we have added the following sentence **on page 8, line 1-9 in the revised manuscript:**

“ To optimize the ion content in HD-ME, we carefully controlled the ionic concentration based on the results from the optical microscopy experiment in Supplementary Fig. 8. When the ion content was set at 50 wt% relative to the chitosan matrix, no ion leakage occurred even under pressure. However, when the ion content was increased to 60 wt%, the material initially formed as a fine film with no ion leakage, whereas ion leakage occurred when pressure was applied. At ion concentrations above 80 wt%, ion leakage was observed even without pressure, and applying pressure further accelerates ion leakage. As a result, we determined that 50 wt% ion content was optimal for minimizing ion leakage while maintaining the desired material properties.”

Supplementary Fig. 8 | Optical microscopy images of HD-ME under different ionic liquid (IL) contents: **a**, 50 wt% IL, **b**, 60 wt% IL, and **c**, 80 wt% IL, before and after pressure application.

Q3) The specific caption 'Mechanotransduction ion dynamics' is somewhat vague for Figure 3c. Since the plot directly displays the loss tangent $\tan \delta$ spectra, it would be more accurate to label it as 'Frequency dependence of dielectric loss $\tan \delta$ under different pressures'. This clearly describes the observable data, while the text can discuss how this reflects the underlying dynamics.

Response: We sincerely appreciate the reviewer's excellent suggestion and have updated the figure caption to "**Frequency dependence of dielectric loss $\tan \delta$ under different pressures.**" This revision more accurately describes the data presented in **Figure 3c**. The plot shows the changes in dielectric loss ($\tan \delta$) across frequencies under varying pressure conditions, which highlights how the material's dielectric properties are influenced by external pressure. We believe this updated caption clearly conveys the information shown in the figure.

Q4) The decrease in relaxation time with increasing pressure in Figure 3d appears to contradict the expectation that compression usually hinders ion transport due to reduced free volume. Please provide a more fundamental explanation: is this result an artifact of reduced contact resistance, or does it reflect a genuine change in the ionic hopping mechanism?

Response: We appreciate the reviewer's insightful comment regarding the decrease in relaxation time with increasing pressure in Figure 3d. We agree that, in conventional polymer

electrolytes or ionic liquids, compression is often associated with hindered ion transport due to reduced free volume. In our HD-ME ion gel system, however, the key design principle is that ions are initially trapped at the ME-functionalized Au nanoparticles via hydrogen bonding, and that applied pressure serves as the driving force to release these trapped ions available to form the electric double layer (EDL). The relaxation time (τ) is inversely related to the free ion density (c_i) and can be described by the following relationship³⁸⁻³⁹:

$$\tau \propto \frac{1}{c_i}$$

where c_i is the concentration of free ions, which is enhanced by pressure-induced release from their binding sites. This increase in free ions accelerates the EDL formation and shortens the relaxation time, as ions are now able to more quickly accumulate at the electrode interfaces. This is directly reflected in the shift of the Debye relaxation frequency to higher values under pressure. In contrast, in systems where ions are not released or where the free ion density does not increase significantly, no such shift in relaxation time is observed, as seen in the pristine ion gel. Thus, the observed decrease in relaxation time with increasing pressure is not an artifact of measurement but a genuine consequence of the increased free ion concentration and the resulting faster ionic dynamics.

We hope this explanation clarifies the relationship between pressure and relaxation time, emphasizing the role of free ion concentration in governing the observed ion dynamics and the overall system behavior under mechanical stimuli. To emphasize this significant advancement, we have included the following sentence **on page 11, line 11-20 in the revised manuscript**:

“To further explain the observed decrease in relaxation time with increasing pressure, we emphasize the relationship between relaxation time and free ion concentration. Relaxation time (τ) is inversely proportional to the concentration of free ions (c_i) in the gel, as given by the equation:

$$\tau \propto \frac{1}{c_i}$$

As pressure is applied, trapped ions within the matrix are released, increasing the concentration of free ions available to form the electric double layer (EDL). This increase in free ions accelerates the ionic response, leading to a decrease in relaxation time. This pressure-induced change in ion dynamics is reflected in the shift of the Debye relaxation frequency to higher frequencies, as observed in Figure 3d.”

[38] Wu, Jianzhong. Understanding the electric double-layer structure, capacitance, and charging dynamics. *Chemical Reviews* 122 10821-10859 (2022)

[39] Singh, Maibam Birla, and Rama Kant. Debye–Falkenhagen dynamics of electric double layer in presence of electrode heterogeneities. *Journal of Electroanalytical Chemistry* 704 197-207 (2013)

Q5) Regarding Figure 3f, I noticed that the 3 dB bandwidth for the CS+IL and HD-ME samples increases significantly with rising pressure, suggesting a degradation in the Quality Factor

(Q). I am interested to know how the Q factor evolves under pressure for these groups. Furthermore, considering this trade-off between spectral sharpness and sensitivity, what is the primary advantage of the HD-ME configuration that justifies its selection as the optimal material?

Response: We appreciate the reviewer's suggestion. Regarding Figure 3f, the observed increase in the 3 dB bandwidth with applied pressure should be interpreted as a change in the shape of the resonant response, rather than a shift in the resonant frequency itself. From a resonant-circuit perspective, the 3 dB bandwidth is directly related to the quality factor (Q), and an increase in the effective resistive component generally leads to a reduced Q and a correspondingly broadened bandwidth. In the present system, the HD-ME configuration exhibits a more pronounced pressure-dependent variation in the equivalent circuit characteristics compared to the other materials. In equivalent-circuit terms, this corresponds to a larger pressure-induced change in the effective resistive and impedance components, which manifests as a clearer modification of the resonance bandwidth and profile in S_{11} -based frequency-sweep measurements. As a result, the resonant responses under different pressure conditions become more distinguishable.

From a wireless sensing and readout standpoint, the key performance metric is not the sharpness of the resonance (i.e., a high Q) alone, but rather how clearly and reproducibly the resonant response characteristics vary with pressure under realistic measurement conditions. Because the HD-ME configuration shows the largest pressure-dependent change in Q and bandwidth among the three materials, the corresponding resonant response patterns are more readily separable across pressure states, which improves pressure discrimination and readout robustness. Therefore, the reduction in Q and the accompanying bandwidth broadening observed for the HD-ME configuration should not be interpreted as a performance degradation, but rather as an indication of enhanced pressure-dependent modulation of the equivalent circuit parameters. For this reason, HD-ME was selected as the most suitable configuration for reliable pressure-state discrimination in the proposed wireless sensing system.

Q6) The schematic illustration of the hydrogen bonding mechanism in Figure 3a is chemically inaccurate. Please correct this to show the appropriate donor-acceptor interaction ($H\cdots O$).

Response: We appreciate the reviewer's suggestion and have updated **Figure 3a** to correct the hydrogen bond depiction. As a result, hydrogen bonds are now represented as donor-acceptor interactions in the chemically correct form $H\cdots O$, where H is the hydrogen bond donor and O is the acceptor.

Fig. 3 | a, Mechanotransduction of the HD-ME iongel, where ions are hydrogen-bonded in the pre-stimulus state and released under applied pressure.

Q7) In Figure 4, there are several figures with no scale bar, such as panel 4a, the upper right picture of 4b, 4e and 4g. The figures contain non-standard notations, such as 'S11', 'cm-1' (fig. 2b), which lack the necessary subscripts and superscripts.

Response: We appreciate the reviewer's careful suggestions. As the reviewer has carefully pointed out, we have revised the figures and labels. Specifically, we added scale bars to all relevant images of **Figure 4 (including images 4a, 4b, 4e, and 4g)** and revised the corresponding figure captions to clearly indicate the scale. We also standardized notation throughout the manuscript to follow common scientific notation, using appropriate subscripts and superscripts. For example, we changed "S11" to S_{11} and "cm-1" to cm^{-1} (**Figure 2b, 3f, 4f and 4h**). We also ensured that all symbols, units, and exponents were consistent throughout all figures and text.

Fig. 2 | b, FTIR spectra of in the spectral regions of 3030-2870 cm^{-1} (S-Au and CH_2 peak) and 2595-2505 cm^{-1} (S-H peak).

Fig. 3 | f, Wireless resonant frequency sweeps at different pressures of the CS+IL iongel and HD-ME iongel, respectively.

Fig. 4 | Demonstration of wireless sensing between normal and oily artery conditions. a, Photographs of the wireless pressure sensor platform integrated with balloon catheter and inflator. **b**, Schematic illustration and photograph of a normal and oily artery model's blood pressure dynamics. **c-d**, Relative capacitance change plots and wireless pressure sensing comparison between normal and oily artery as a types of ion gel. Inset shows the capacitance changes and resonance frequency changes between artery expansion and constriction. **e**, Photograph of the balloon catheter integrated with WiLECS, incorporating a spacer between the RX and TX units. **f**, Resonance frequency shifts as a function of catheter pressure, measured in a configuration with a spacer inserted between the TX and RX antennas. **g**, Photograph of the balloon catheter integrated with WiLECS, incorporating a porcine skin between the RX and TX units. **h**, Resonance frequency shifts as a function of catheter pressure, measured in a configuration with a porcine skin inserted between the TX and RX antennas.

Q8) While the manuscript notes low-frequency stability, Figure S15 shows capacitance drift at 30 kPa. Please define the specific stable pressure range and clarify if the wireless frequency readout exhibits similar drift under this pressure.

Response: We sincerely appreciate the reviewer's valuable and insightful question. As the reviewer correctly noted, ion drift can occur at pressures exceeding 30 kPa, as shown in Figure S15. This behavior is closely related to the fundamental characteristics of ion transport in low-conductivity ionic systems. Specifically, the observed drift arises when the timescale of the wireless readout, governed by the effective ionic mobility, is slower than the redistribution of

ions within the material. This effect is particularly relevant in our system because we intentionally employed choline-malate ionic liquid to enhance material stability. The formation of ion clusters in this ionic liquid reduces ionic mobility, which in turn leads to slower dynamic responses and makes drift observable under high-pressure conditions.

In principle, a similar drift would also appear in the wireless frequency readout if capacitance drift were to occur, because the capacitance directly determines the resonance frequency of the LC circuit. However, it is important to emphasize that the pressure level at which this drift becomes noticeable (~30 kPa) is far beyond the range encountered in typical physiological scenarios, for example, human pulse or blood-pressure-related deformations are generally well below 1 kPa. Therefore, the observed drift represents an extremely upper operating limit rather than a practical sensing condition. Crucially, within the pressure range relevant to physiological monitoring, the system remains stable. As shown in Figure 4d of the main manuscript, no measurable shift in resonance frequency is observed during sensing, indicating robust and drift-free operation. Based on these results, we are confident that the device performs reliably under realistic operating conditions and is well suited for real-world physiological pressure monitoring applications.

Q9) Real blood vessels are composed of multilayered biological tissues within an ion-rich physiological environment, which differs fundamentally from the synthetic model. This discrepancy may introduce background noise or signal attenuation, thereby affecting the SNR. To ensure clinical relevance, the authors are advised to validate the device performance using real animal blood vessels.

Response: We sincerely thank the reviewer for this insightful and constructive comment. We fully agree that real blood vessels consist of multiple biological tissue layers and operate in an ion-rich physiological environment, which is inherently more complex than our artificial artery model. Such complexity can introduce additional background noise or signal distortion and may potentially affect the signal-to-noise ratio (SNR). To directly address this concern, we performed additional validation experiments in response to the reviewer's comment by operating the device under conditions that more closely approximate the human physiological environment. Specifically, we conducted wireless sensing measurements in phosphate-buffered saline (PBS) at 37 °C, thereby reproducing both the ionic composition and thermal conditions relevant to in vivo operation. This additional experiment enabled us to explicitly evaluate the influence of ionic screening and hydration on the wireless readout and SNR.

We employed a balloon catheter model rather than real animal blood vessels because its diameter (~6 mm) and mechanical modulus (on the order of several MPa)¹ are among the closest available surrogates to human arteries. Although animal vessels (e.g., rat or rabbit arteries) could be considered, their substantially smaller diameters and distinct wall mechanics deviate significantly from those of human vasculature. Given the practical and ethical limitations associated with direct human testing, we believe that this catheter-based system provides a more relevant and well-controlled biomechanical and geometrical proxy for evaluating human blood-pressure sensing performance. As anticipated by the reviewer, the ion-

and water-rich PBS environment resulted in a modest reduction in SNR compared to simplified conditions. Nevertheless, we observed stable and clearly distinguishable blood-pressure sensing without loss of functionality, demonstrating that the device maintains reliable wireless operation under physiologically relevant conditions.

We appreciate the reviewer's suggestion, which motivated these additional validation experiments and allowed us to more clearly articulate both the physiological relevance and the limitations of our experimental model. To improve transparency and directly address this point, the new data has been included as **Supplementary Fig. 22**. We have also added a corresponding statement to the main text, along with a clarification of the SNR measurement methodology, to strengthen the rationale underlying our interpretation.

Main section (Demonstration of WiLECS in artificial artery models for wireless blood pressure monitoring) on page 16 and 17

“To further assess the practical reliability of the WiLECS platform under physiologically relevant conditions, we evaluated its sensing performance in an ion-rich aqueous environment. Specifically, the device was operated in phosphate-buffered saline (PBS) at 37 °C to emulate the ionic composition and thermal conditions of biological tissues (Supplementary Fig. 22). In terms of conductivity, which verify the behavior of ions, for the first 10 hours, ion conductivity increased due to the infusion of water and ions. However, it was confirmed that the gel formed stably even in PBS solution with minimal ion leakage, as it exhibited constant ion conductivity thereafter. Compared to measurements conducted under ambient conditions, a modest reduction in sensitivity and SNR was observed, which can be attributed to increased ionic screening and hydration effects. Nevertheless, the WiLECS maintained stable wireless operation and clear pulse-resolved signal readout, indicating that the overall sensing fidelity and functional trend remain robust in physiologically relevant environments.”

Method section (Analysis of pressure response and SNR) on page 22

“The signal-to-noise ratio (SNR) was quantified from the pressure time-series using a peak-to-peak ($p-p$) definition for the signal and a root-mean-square (RMS) definition for the noise. The signal amplitude $V_{pp,signal}$ was defined as the difference between the maximum and minimum values within a representative pulse window. The noise level (σ_{noise}) defined as the standard deviation (RMS) of the baseline segment acquired under no pulsation.”

Supplementary Fig.22 | Signal-to-noise ratio (SNR) of WiLECS under physiologically relevant conditions. a, Photograph of the WiLECS in PBS solution b, Multi-days relative conductivity change plot for ion leaching test c, Relative capacitance change plots of WiLECS in ambient condition d, Relative capacitance change plots of WiLECS in PBS condition after 15days

[1] *The Journal of thoracic and cardiovascular surgery* 142.3 (2011): 682-686.

Q10) Please specify the exact pressure applied to achieve the 78 dB SNR in Figure S20. Providing the corresponding frequency signal and noise floor data would greatly enhance the completeness and transparency of this characterization.

Response: We sincerely appreciate the reviewer's thoughtful and valuable comment. The SNR data presented in Supplementary Fig. 20 were obtained from our blood pressure artery model, which closely mimics physiological conditions. Specifically, the 78 dB SNR corresponds to a pressure of approximately 5 kPa, which was applied within the artificial artery during the experiment. We fully understand the importance of providing a more comprehensive and transparent characterization, and we agree that including the corresponding frequency signal and noise floor data would enhance the completeness of this measurement. To this end, we plan to include all raw data associated with these measurements in the final submission, ensuring that all details are provided for full reproducibility. Once again, we greatly appreciate the reviewer's input, which has helped improve the clarity and transparency of the manuscript. To highlight this noteworthy progress, we have incorporated the following sentence **on page 15 in the revised manuscript:**

"The SNR value was calculated based on the pressure data obtained from our previously

established artificial artery system. Specifically, the pressure of approximately 5 kPa, which was generated within the system, served as the basis for determining the SNR. This approach allowed us to closely replicate physiological conditions and accurately assess the SNR under realistic arterial pressure scenarios.

Reply to Reviewer #2

Thank you for your invaluable comments. We fully revised the manuscript according to your comments.

Overall Comment: *This manuscript presents a soft, wireless LC pressure sensor (WiLECS) in which a mechanosensitive ion gel converts arterial pressure into large, low-frequency capacitance modulation through a hydrogen-bond-mediated trap-and-release mechanism. The mechanisms of mechanically gated ionic trapping/release and resonance-frequency tuning are elucidated through systematic analyses, along with demonstrations of wireless LC behavior in arterial phantom models. This is a highly relevant and compelling topic in the fields of ionotronics and biointegrated cardiovascular monitoring, and the work could provide an even deeper mechanistic understanding for researchers and readers. I recommend that the manuscript be considered for publication in Nature Communications after minor revisions addressing the detailed comments below.*

Response: We sincerely thank Reviewer 2 for the positive assessment and for recognizing the novelty of our work. We have carefully addressed all comments, and these revisions have significantly improved the quality of the original manuscript. We hope that Reviewer 2 finds the revised version satisfactory.

Q1) *The main concept appears to be a wireless LC pressure sensor with an ionic gel capacitor, Science Advances 11(11) (2025), eadu6086, has demonstrated polyelectrolyte-based wireless iontronic sensors using an ionic medium. Could the authors more clearly clarify the conceptual and practical novelty of WiLECS relative to both conventional elastomeric LC sensors and this recent ionic/iontronic work, and also include the device from Sci. Adv. 11(11), eadu6086 in the Figure. If figure-of-merit comparison for a quantitative benchmark?*

Response: We appreciate the reviewer's suggestion. We have carefully reviewed the cited work and have added the *Science Advances* paper to the figure-of-merit comparison in **Figure 1f** to enable a more direct and quantitative benchmark. The referenced *Science Advances* study integrates a polyelectrolyte elastomer-based iontronic capacitor with an LC resonant circuit, with the primary objective of minimizing creep-induced drift for real-time health monitoring. In that work, the iontronic LC sensors are mainly benchmarked for wireless readout in the high-frequency range of 100 MHz to 2 GHz. In contrast, our system is specifically designed to operate continuously in the low-frequency regime below 1 MHz, which is advantageous for biological compatibility and robust operation in physiological environments. Moreover,

WiLECS fundamentally differs from conventional iontronic approaches that rely on pressure-dependent changes in contact area. Instead, WiLECS actively modulates the local dielectric environment of the resonant circuit through piezo-driven ion redistribution, enabling high-sensitivity pressure sensing at low frequencies. As a result, the system achieves a sensitivity of 0.0036 mmHg^{-1} at low resonance frequencies. In addition, our work goes beyond demonstrating an isolated sensing element by integrating an antenna into the device architecture, enabling stable and reliable wireless operation. Consequently, WiLECS supports stable sub-MHz wireless pressure sensing, as demonstrated by real-time, wireless tracking of blood-pressure signals under clinically relevant conditions.

Fig. 1 | f, Comparison of the previously reported sensitivity and operating frequency range for wireless pressure sensors.

Q2) For practical use in wearable or implantable settings, environmental stability (thermal and electrochemical) is critically important for reliable device operation. I therefore encourage the authors to more explicitly address the stability of the device/gel under relevant conditions (e.g., body temperature, and long-term bias etc), or at least to discuss these aspects and the expected failure modes and limitations.

Response: We sincerely thank the reviewer for raising this important point. We agree that thermal and electrochemical stability under physiologically relevant conditions is essential for wearable and implantable use, and we have added additional data and discussion to address this more explicitly. To evaluate stability in an ion-rich environment at near-body temperature, we operated the gel/device in PBS at $37 \text{ }^\circ\text{C}$ and tracked ionic conductivity as an indicator of ion dynamics and potential leaching. The conductivity increased during the first ~ 10 hours due to hydration and ionic equilibration, then reached a steady value and remained constant thereafter, indicating stable gel formation in PBS with minimal IL leakage. We also tested gels after 15 days of PBS immersion in a PBS-based artificial artery model. While sensitivity and SNR showed a modest decrease compared with ambient measurements, the WiLECS maintained stable wireless operation and clear pulse-resolved readout, supporting robust functionality after prolonged exposure to physiologically relevant media (**Supplementary Fig. 22**). In addition, we assessed thermal safety during wireless operation using infrared thermal

imaging. The device temperatures measured before, during, and after operation were 19.1 °C, 21.8 °C, and 22.4 °C, respectively, showing no operation-induced temperature rise (**Supplementary Fig. 23**). To highlight this important progress, we have added the following sentence on page 16, 17 and supplementary figure 22, 23 in the revised manuscript:

“To further assess the practical reliability of the WiLECS platform under physiologically relevant conditions, we evaluated its sensing performance in an ion-rich aqueous environment. Specifically, the device was operated in phosphate-buffered saline (PBS) at 37 °C to emulate the ionic composition and thermal conditions of biological tissues (Supplementary Fig. 22). In terms of conductivity, which verify the behavior of ions, for the first 10 hours, ion conductivity increased due to the infusion of water and ions. However, it was confirmed that the gel formed stably even in PBS solution with minimal ion leakage, as it exhibited constant ion conductivity thereafter. Compared to measurements conducted under ambient conditions, a modest reduction in sensitivity and SNR was observed, which can be attributed to increased ionic screening and hydration effects. Nevertheless, the WiLECS maintained stable wireless operation and clear pulse-resolved signal readout, indicating that the overall sensing fidelity and functional trend remain robust in physiologically relevant environments.”

Supplementary Fig.22 | Signal-to-noise ratio (SNR) of WiLECS under physiologically relevant conditions. a, Photograph of the WiLECS in PBS solution **b**, Multi-days relative conductivity change plot for ion leaching test **c**, Relative capacitance change plots of WiLECS in ambient condition **d**, Relative capacitance change plots of WiLECS in PBS condition after 15days

“In addition, to evaluate the bio-safety of sub-MHz wireless operation, we monitored the device temperature using infrared thermal imaging before, during, and after 6 hours operation. The measured temperatures were 19.1 °C, 21.8 °C, and 22.4 °C, respectively, with no significant temperature rise observed, indicating negligible Joule heating under the operating conditions (Supplementary Fig. 23).”

Supplementary Fig.23 | Thermal characterization of the WiLECS during wireless operation. Infrared thermal images of the WiLECS device recorded before **a**, wireless operation, **b**, during steady-state operation, and **c**, after operation, measured using a thermal imaging camera under the same operating conditions as the wireless sensing experiments.

Q3) The authors attribute the enhanced sensitivity to a “trap-and-release” mechanism at ME-functionalized AuNPs. However, the observed FT-IR shifts and relaxation-time changes could also be interpreted as generic pressure-induced changes in ion distribution. Can the authors more convincingly argue that ion trapping at AuNPs is essential, rather than simple EDL compression?

Response: We appreciate this thoughtful mechanistic question. We agree that pressure inevitably perturbs the EDL even in conventional ion gels, and we have revised the text to more clearly distinguish the baseline EDL compression from the additional trap-and-release process. Mechanistically, three observations support the essential role of ion trapping at ME-functionalized AuNPs:

1. **Baseline capacitance:** HD-ME ion gels exhibit significantly lower initial capacitance than CS+IL gels with the same IL content, despite having similar geometry. This is consistent with a reduced population of mobile ions because a fraction is immobilized at the AuNP surface through hydrogen bonding.
2. **Field dependence:** In MIM devices, increasing the AC bias strongly increases capacitance for CS+IL gels, but has only a minor effect in HD-ME gels, indicating that trapped ions cannot be freely mobilized by the electric field alone.
3. **Pressure dependence:** Under pressure, HD-ME gels show both a larger change in relaxation time and a much larger $\Delta C/C_0$ than CS+IL, even though both are subjected to the same mechanical and electrical conditions. The FTIR peak shifts in the C–O region further support the partial disruption of hydrogen bonds around choline and malate upon compression.

We have clarified this argument in the “*Multiscale characterization of ion trapping and restricted capacitance dynamics*” section, emphasizing that the enhanced pressure sensitivity arises from the combination of trapped ions at rest and their mechanically triggered release, rather than EDL compression alone. No additional experiments were required; instead, we re-organized the existing data and text to make the causal chain more explicit.

Q4) In the current manuscript, the main stated advantage of using Au nanoparticles is their biocompatibility, but the specific benefits of Au compared with other possible fillers (e.g., in terms of interfacial chemistry, ionic trapping, mechanical properties, or electrical performance) are not clearly articulated. I encourage the authors to clarify why Au was chosen and what unique role it plays in the gel design beyond biocompatibility.

Response: We thank the reviewer for raising this important and practical consideration. In our system, the Au nanoparticles (AuNPs) are not employed as passive fillers; rather, they function as chemically well-defined, high-surface-area platforms that enable thiol coordination and hydrogen-bond-mediated ion trapping. Gold was selected because it offers several key advantages: (i) strong and stable S–Au bonding with ME, (ii) excellent chemical stability under physiological and electrochemical conditions, and (iii) well-established surface chemistry that allows systematic tuning of the surface functionalization density through controlled variation of pH and particle size. Although other inorganic fillers could, in principle, be employed, our primary objective in this first demonstration was to establish a clear and unambiguous structure-mechanism relationship using a robust and well-controlled interfacial system. We **have now explicitly clarified this rationale in the revised manuscript**. We also note that substituting AuNPs with lower-cost alternative materials represents an interesting and important direction for future optimization; however, this lies beyond the scope of the present mechanistic study.

Reply to Reviewer #3

Thank you for your invaluable comments. We fully revised the manuscript according to your comments.

Overall Comment: The article entitled “Low-frequency ionic-electronic coupling for energy-efficient noise-resilient wireless bioelectronics” by Kim et al. introduces an approach for wireless electro-chemical sensing that combines ionic dynamics with low-frequency LC circuits. The proposed material and concept would be useful for the development of associated technologies, but some of the below comments should be addressed.

Response: We sincerely thank Reviewer 3 for the positive assessment and for recognizing the novelty of our work. We have carefully addressed all comments, and these revisions have significantly improved the quality of the original manuscript. We hope that Reviewer 3 finds the revised version satisfactory.

Q1) Explain the selection of 50 wt.% IL as the primary composition (Authors claim that >50% causes leaching). To support the choice of 50 weight percent, provide a brief analysis of the trade-off IL% vs. modulus vs. capacitance vs. leaching.

Response: We sincerely appreciate the reviewer for this constructive suggestion. As shown in Supplementary Figs. 3 and 4, increasing the ionic liquid (IL) content in the chitosan matrix systematically increases both the mechanical softness (reduced modulus) and the capacitance, consistent with the plasticization effect and the higher density of mobile ions contributing to interfacial polarization/EDL formation. Importantly, this trend is expected to continue beyond 50 wt% IL as well. However, above 50 wt% IL, the material response becomes increasingly dominated by instability associated with ionic liquid leaching and non-uniform film formation. In this regime, the measured mechanical and electrical properties can no longer be interpreted as intrinsic “bulk gel” properties, because the effective IL fraction in the remaining matrix changes dynamically during handling and under pressure, and the device response can be confounded by interfacial contamination and time-dependent leakage. For this reason, we did not include modulus/capacitance values above 50 wt% in the main analysis, as doing so could be misleading by implying stable gel behavior when the composition is not mechanically or chemically retained. To provide a more direct and intuitive validation of this trade-off and to support our claim of 50 wt% IL as the primary composition, we added optical microscopy observations of HD-ME ionogels at 50, 60, and 80 wt% IL (**Supplementary Figure 8**). At 50 wt% IL, no visible IL leakage was observed even under compression, indicating a stable matrix-IL compatibility suitable for pressure sensing. At 60 wt% IL, the film initially appeared uniform without apparent leakage however, IL exudation was clearly triggered under applying pressure, demonstrating that the composition is mechanically unstable under sensing operating condition. At 80 wt% IL, IL leakage was observed even without pressure, struggling stable gel formation and reliable device operation.

Overall, higher IL increases capacitance and softens the gel, but above 50 wt% it causes leaching and unstable mechanics. We therefore chose 50 wt% IL as the highest stable composition without pressure-induced leakage. To highlight this important progress, we have added the following sentence **on page 8, line 1-9 in the revised manuscript**:

" To optimize the ion content in HD-ME, we carefully controlled the ionic concentration based on the results from the optical microscopy experiment in Supplementary Fig. 8. When the ion content was set at 50 wt% relative to the chitosan matrix, no ion leakage occurred even under pressure. However, when the ion content was increased to 60 wt%, the material initially formed as a fine film with no ion leakage, whereas ion leakage occurred when pressure was applied. At ion concentrations above 80 wt%, ion leakage was observed even without pressure, and applying pressure further accelerates ion leakage. As a result, we determined that 50 wt% ion content was optimal for minimizing ion leakage while maintaining the desired material properties."

Supplementary Fig. 8 | Optical microscopy images of HD-ME under different ionic liquid (IL) contents: a. 50 wt% IL, b. 60 wt% IL, and c. 80 wt% IL, before and after pressure application.

Q2) The sample names (HD-ME, LD-ME, CS+IL, pristine CS) aren't always used consistently throughout the paper. Please keep the name uniform throughout to avoid confusion).

Response: We are grateful to the reviewers for their careful and perceptive assessment of our manuscript, as well as for their valuable and constructive feedback. After careful review of the manuscript, we adopted a unified sample nomenclature to improve clarity and consistency. We believe these revisions will help readers understand the manuscript more clearly and avoid confusion. Accordingly, the sample names used throughout the manuscript have been revised as follows:

- *chitosan (CS)-based ion gels* → **chitosan-based ion gels (CS+IL ion gel)** on page 5,
- *pristine ion gel (CS+IL, 50 wt% IL)* → **CS+IL ion gel (50 wt% IL)** on page 7,
- *CS ion gel* → **CS+IL ion gel** on page 8,
- *The HD-ME-based* → **HD-ME ion gel-based** on page 14,
- *The HD-ME-based* → **HD-ME ion gel-based** on page 16,
- *CS polymer gel-based* → **CS gel-based** on page 16.

Q3) Fabricate identical gels with AuNPs that are not functionalized (pristine AuNP) and with ME alone (no AuNP) to show the necessity of both ME-AuNP coordination and AuNP mechanical contrast for the trap-release behavior. Some comparisons exist (they show AuNP

and LD-ME), but a complete control set (no AuNP, pristine AuNP, ME-only) would be helpful.

Response: We appreciate the reviewer's insightful and constructive comment. We agree that a complete set of control experiments is essential to clearly demonstrate the necessity of both ME-AuNP coordination and the mechanical contrast introduced by AuNPs for the proposed trap-release mechanism. In response, we performed additional comparative experiments using chitosan-based ion gels with the following control compositions:

- (i) a pristine chitosan ion gel without any additives,
- (ii) Ion gel containing pristine (non-functionalized) AuNPs,
- (iii) Ion gel containing ME alone without AuNPs.

The pressure sensitivities of these control samples were measured under identical conditions and compared with that of the HD-ME ion gel. The measured sensitivities were 0.237 kPa^{-1} for the pristine chitosan gel, 0.745 kPa^{-1} for the gel containing pristine AuNPs, and 0.0938 kPa^{-1} for the ME-only gel. In contrast, the HD-ME ion gel exhibited a much higher sensitivity of 17.36 kPa^{-1} . This pronounced difference indicates that neither the presence of AuNPs alone nor ME alone is sufficient to induce strong pressure-responsive behavior. Instead, the combination of surface-functionalized AuNPs and their coordinated interaction with ME is required to enable effective ion trapping at rest and pressure-induced ion release. These results support our interpretation that only ME-functionalized AuNPs can generate pressure-sensitive ion release behavior via stress concentration and the resulting von Mises stress at the particle-matrix interface, leading to a large modulation of free-ion population under compression. We have added control data and the corresponding discussion to the revised manuscript to clarify the necessity of both components in realizing the proposed trap-release mechanism. To address this point, we have included the pressure-responsive behavior data corresponding to the different additives discussed above in **Supplementary Fig. 15**.

Supplementary Fig. 15 | Comparison of capacitance change as a function of applied pressure. The capacitance change was analyzed by different functionalized condition (a) Different particle condition, (b) gold nanoparticle size (50-200nm), and (c) synthesizing pH condition (pH 3.8-pH 9.1). The sensitivity of capacitance changes to pressure increases with higher surface functionalization density.

Q4) Given that the sensing mechanism depends on the trapping and release of ions from the AuNP surfaces, it would be crucial to determine the stability of this behavior over time. The SI shows short-term durability and 48-hour cell viability, but there's no information on ion or IL

leaching, or how the device performs over several days or weeks in physiological conditions (37 °C saline). Since these are crucial for evaluating implantable use, I suggest including accelerated leaching tests and multi-day wireless stability measurements (baseline drift and sensitivity after few days gap).

Response: We sincerely thank the reviewer for this important suggestion. We fully agree that, because our sensing mechanism relies on the trap-and-release behavior of ions at the AuNPs interfaces, it is essential to assess whether the ion/IL environment remains stable over time under physiological conditions. To address this concern, we performed additional long-term stability tests of the iongel in aqueous, ion-rich environments. Specifically, we monitored gel behavior in water and PBS, focusing on ionic conductivity as an indicator of ion dynamics and potential leaching. During the first ~10 hours, the ionic conductivity increased, which we attribute to initial hydration and the infusion of water/ions into the gel. Importantly, after this equilibration period, the ionic conductivity became constant, indicating that the gel reaches a stable state in PBS with minimal ion/IL loss thereafter.

In addition, to evaluate whether this stabilized gel state translates into practical device reliability, we immersed the gel in PBS for 15 days and then tested it in our PBS-based artificial artery model. After this prolonged immersion, we observed a modest decrease in sensitivity and SNR compared to ambient measurements, which is consistent with ionic screening and hydration effects in physiological media. Nevertheless, the device still exhibited stable operation and clear signal readout, demonstrating that the overall sensing functionality and trend are preserved even after multi-day exposure to PBS. To improve transparency for readers, we have added these results and the corresponding discussion **in the revised manuscript (Supplementary Fig. 22 and following text)**, clarifying the time-dependent stabilization behavior and the resulting multi-day operational reliability under physiologically relevant conditions.

Main section (Demonstration of WiLECS in artificial artery models for wireless blood pressure monitoring) on page 16

“To further assess the practical reliability of the WiLECS platform under physiologically relevant conditions, we evaluated its sensing performance in an ion-rich aqueous environment. Specifically, the device was operated in phosphate-buffered saline (PBS) at 37 °C to emulate the ionic composition and thermal conditions of biological tissues (Supplementary Fig. 22). In terms of conductivity, which verify the behavior of ions, for the first 10 hours, ion conductivity increased due to the infusion of water and ions. However, it was confirmed that the gel formed stably even in PBS solution with minimal ion leakage, as it exhibited constant ion conductivity thereafter. Compared to measurements conducted under ambient conditions, a modest reduction in sensitivity and SNR was observed, which can be attributed to increased ionic screening and hydration effects. Nevertheless, the WiLECS maintained stable wireless operation and clear pulse-resolved signal readout, indicating that the overall sensing fidelity and functional trend remain robust in physiologically relevant environments.”

Supplementary Fig.22 | Signal-to-noise ratio (SNR) of WiLECS under physiologically relevant conditions. a, Photograph of the WiLECS in PBS solution **b**, Multi-days relative conductivity change plot for ion leaching test **c**, Relative capacitance change plots of WiLECS in ambient condition **d**, Relative capacitance change plots of WiLECS in PBS condition after 15days

Q5) The main idea of the paper is that higher sensitivity and SNR result from starting with a low initial capacitance from trapped ions and then obtaining a large ΔC under pressure. To help readers fully see this relationship, it would be useful to include a simple quantitative comparison, maybe a table showing the initial C , ΔC , Δf , and SNR for each material measured under the same conditions.

Response: We appreciate the reviewer's thoughtful suggestion and believe that a concise quantitative summary will help readers more clearly understand our core premise, namely that a low initial capacitance arising from ion trapping, combined with a large pressure-induced capacitance change, leads to enhanced sensitivity and signal-to-noise ratio. In response to this suggestion, we have added comparative data in **Supplementary Table 4**, where key parameters such as the initial capacitance, normalized capacitance change, resonance frequency shift, and SNR are summarized for each material measured under identical conditions with a pressure range of 0 to 50 kPa. We believe that this addition provides a clearer basis for comparison and improves the overall readability of the manuscript.

Supplementary Table. 4 | Quantitative Comparison of Initial Capacitance, Pressure-Induced Capacitance Modulation, Resonance Shift, and SNR Across Different Gel Capacitor Material ($\Delta P = 0\text{-}50$ kPa).

	CS gel	CS+IL ion gel	HD-ME ion gel
Initial C ($\mu\text{F}/\text{cm}^2$)	3.75×10^{-2}	34.203	0.59663
$\Delta C/C_0$	0.20994	6.3019	392.82189
Δf (kHz)	-7.313	-33.29	-192.6
SNR (dB)	9.3023	31.121	60.94
Sensitivity (mmHg^{-1})	5.60×10^{-4}	0.0168	1.047

Q6) *The manuscript highlights the advantage of using sub-MHz frequencies for bio-safe operation, but actual safety depends on factors like specific absorption rate, induced electric fields, and any tissue heating. Including some measurements or simulations that illustrate the field strength or temperature change for your operating conditions and distances would be very beneficial. If the SNR is actually very low, a brief explanation of safe operating limits would be helpful; if not, data would support the claim.*

Response: We thank the reviewer for this thoughtful comment regarding the biosafety of sub-MHz operation. We fully agree that, beyond frequency selection, practical safety considerations should account for potential induced fields, Joule heating, and temperature rise in surrounding media. To address this concern, we performed direct thermal characterization of the WiLECS during wireless operation using an infrared thermal imaging camera. The device temperature was measured before operation, during steady-state wireless operation, and after operation under the same conditions used for sensing experiments. The recorded temperatures were 19.1 °C (before), 21.8 °C (during), and 22.4 °C (after 6 hours operating), showing no abrupt or excessive temperature increase associated with device operation. The observed temperature change remained within a narrow range consistent with ambient fluctuation and did not indicate appreciable Joule heating or thermal loading. These results provide experimental evidence that the operating conditions of the WiLECS do not induce significant temperature rise, supporting the bio-safe nature of the sub-MHz wireless readout scheme. We have added these data and a corresponding discussion to the revised manuscript to clarify the thermal safety aspect of device operation. To highlight this important progress, we have added the following sentence **on page 17 and supplementary Figure 23 in the revised manuscript:**

“In addition, to evaluate the biosafety of sub-MHz wireless operation, we monitored the device temperature using infrared thermal imaging before, during, and after 6 hours operation. The measured temperatures were 19.1 °C, 21.8 °C, and 22.4 °C, respectively, with no significant temperature rise observed, indicating negligible Joule heating under the operating conditions (Supplementary Fig. 23).”

Supplementary Fig.23 | Thermal characterization of the WiLECS during wireless operation. Infrared thermal images of the WiLECS device recorded before **a**, wireless operation, **b**, during steady-state operation, and **c**, after operation, measured using a thermal imaging camera under the same operating conditions as the wireless sensing experiments.